# QUANTIZED COMPRESSED SENSING WITH SCORE-BASED GENERATIVE MODELS

Xiangming Meng and Yoshiyuki Kabashima

Institute for Physics of Intelligence and Department of Physics
The University of Tokyo
7-3-1, Hongo, Tokyo 113-0033, Japan
{meng,kaba}@g.ecc.u-tokyo.ac.jp

## ABSTRACT

We consider the general problem of recovering a high-dimensional signal from noisy quantized measurements. Quantization, especially coarse quantization such as 1-bit sign measurements, leads to severe information loss and thus a good prior knowledge of the unknown signal is helpful for accurate recovery. Motivated by the power of score-based generative models (SGM, also known as diffusion models) in capturing the rich structure of natural signals beyond simple sparsity, we propose an unsupervised data-driven approach called quantized compressed sensing with SGM (QCS-SGM), where the prior distribution is modeled by a pre-trained SGM. To perform posterior sampling, an annealed pseudo-likelihood score called *noise perturbed pseudo-likelihood score* is introduced and combined with the prior score of SGM. The proposed QCS-SGM applies to an arbitrary number of quantization bits. Experiments on a variety of baseline datasets demonstrate that the proposed QCS-SGM significantly outperforms existing state-of-the-art algorithms by a large margin for both in-distribution and out-of-distribution samples. Moreover, as a posterior sampling method, QCS-SGM can be easily used to obtain confidence intervals or uncertainty estimates of the reconstructed results. *The code is available at https://github.com/mengxiangming/QCS-SGM.*

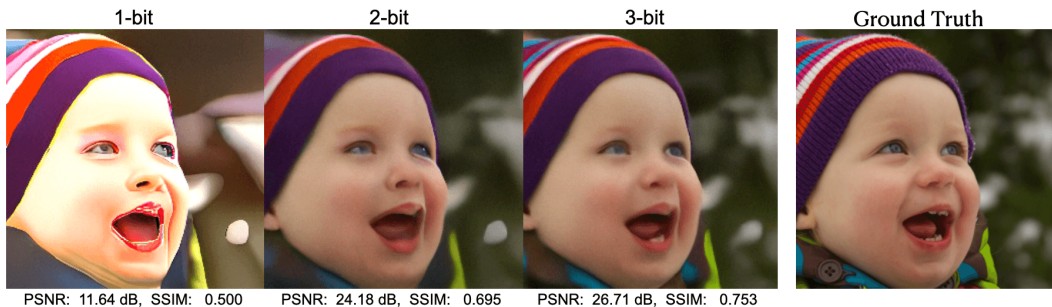

Figure 1: Reconstructed images of our QCS-SGM for one FFHQ $256 \times 256$ high-resolution RGB test image ($N = 256 \times 256 \times 3 = 196608$ pixels) from noisy heavily quantized (1bit, 2-bit and 3-bit) CS $8\times$ measurements $\mathbf{y} = \mathrm{Q}(\mathbf{Ax} + \mathbf{n})$, i.e., $M = 24576 \ll N$. The measurement matrix $\mathbf{A} \in \mathbb{R}^{M \times N}$ is i.i.d. Gaussian, i.e., $A_{ij} \sim \mathcal{N}(0, \frac{1}{M})$, and a Gaussian noise $\mathbf{n}$ is added with standard deviation $\sigma = 10^{-3}$.

## 1 INTRODUCTION

Many problems in science and engineering such as signal processing, computer vision, machine learning, and statistics can be cast as linear inverse problems:

$$\mathbf{y} = \mathbf{Ax} + \mathbf{n}, \tag{1}$$

where $\mathbf{A} \in \mathbb{R}^{M \times N}$ is a known linear mixing matrix, $\mathbf{n} \sim \mathcal{N}(\mathbf{n}; 0, \sigma^2 \mathbf{I})$ is an i.i.d. additive Gaussian noise, and the goal is to recover the unknown signal $\mathbf{x} \in \mathbb{R}^{N \times 1}$ from the noisy linear measurements

$\mathbf{y} \in \mathbb{R}^{M \times 1}$. Among various applications, compressed sensing (CS) provides a highly efficient paradigm that makes it possible to recover a high-dimensional signal from a far smaller number of $M \ll N$ measurements (Candès & Wakin, 2008). The underlying wisdom of CS is to leverage the intrinsic structure of the unknown signal $\mathbf{x}$ to aid the recovery. One of the most widely used structures is *sparsity*, i.e., most elements of $\mathbf{x}$ are zero under certain transform domains, e.g., wavelet and Fourier transformation (Candès & Wakin, 2008). In other words, the standard CS exploits the fact that many natural signals in real-world applications are (approximately) sparse. This direction has spurred a hugely active field of research during the last two decades, including efficient algorithm design (Tibshirani, 1996; Beck & Teboulle, 2009; Tropp & Wright, 2010; Kabashima, 2003; Donoho et al., 2009), theoretical analysis (Candès et al., 2006; Donoho, 2006; Kabashima et al., 2009; Bach et al., 2012), as well as all kinds of applications (Lustig et al., 2007; 2008), just to name a few.

Despite their remarkable success, traditional CS is still limited by the achievable rates since the sparsity assumptions, whether being naive sparsity or block sparsity (Duarte & Eldar, 2011), are still too simple to capture the complex and rich structure of natural signals. For example, most natural signals are not strictly sparse even in the specified transform domains, and thus simply relying on sparsity alone for reconstruction could lead to inaccurate results. Indeed, researchers have proposed to combine sparsity with additional structure assumptions, such as low-rank assumption (Fazel et al., 2008; Foygel & Mackey, 2014), total variation (Candès et al., 2006; Tang et al., 2009), to further improve the reconstruction performances. Nevertheless, these hand-crafted priors can only apply, if not exactly, to a very limited range of signals and are difficult to generalize to other cases. To address this problem, driven by the success of generative models (Goodfellow et al., 2014; Kingma & Welling, 2013; Rezende & Mohamed, 2015), there has been a surge of interest in developing CS methods with data-driven priors (Bora et al., 2017; Hand & Joshi, 2019; Asim et al., 2020; Pan et al., 2021). The basic idea is that, instead of relying on hand-crafted sparsity, the prior structure of the unknown signal is learned through a generative model, such as VAE (Kingma & Welling, 2013) or GAN (Goodfellow et al., 2014). Notably, the past few years have witnessed a remarkable success of one new family of probabilistic generative models called diffusion models, in particular, score matching with Langevin dynamics (SMLD) (Song & Ermon, 2019; 2020) and denoising diffusion probabilistic modeling (DDPM) (Ho et al., 2020; Nichol & Dhariwal, 2021), which have proven extremely effective and even outperform the state-of-the-art (SOTA) GAN (Goodfellow et al., 2014) and VAE (Kingma & Welling, 2013) in density estimation and generation of various natural sources. As both DDPM and SMLD estimate, implicitly or explicitly, the *score* (i.e., the gradient of the log probability density w.r.t. to data), they are also referred to together as score-based generative models (SGM) (Song et al., 2020). Several CS methods with SGM have been proposed very recently (Jalal et al., 2021a;b; Kawar et al., 2021; 2022; Chung et al., 2022) and perform quite well in recovering $\mathbf{x}$ with only a few linear measurements in (1).

Nevertheless, the linear model (1) ideally assumes that the measurements have infinite precision, which is not the case in realistic acquisition scenarios. In practice, the obtained measurements have to be quantized to a finite number of $Q$ bits before transmission and/or storage (Zymnis et al., 2009; Dai & Milenkovic, 2011). Quantization leads to information loss which makes the recovery particularly challenging. For moderate and high quantization resolutions, i.e., $Q$ is large, the quantization impact is usually modeled as a mere additive Gaussian noise whose variance is determined by quantization distortion (Dai & Milenkovic, 2011; Jacques et al., 2010). Subsequently, most CS algorithms originally designed for the linear model (1) can then be applied with some modifications. However, such an approach is apparently suboptimal since the information about the quantizer is not utilized to a full extent (Dai & Milenkovic, 2011). This is true especially in the case of coarse quantization, i.e., $Q$ is small. An extreme and important case of coarse quantization is *1-bit* quantization, where $Q = 1$ and only the *sign* of the measurements are observed (Boufounos & Baraniuk, 2008). Apart from extremely low cost in storage, 1-bit quantization is particularly appealing in hardware implementations and has also proven robust to both nonlinear distortions and dynamic range issues (Boufounos & Baraniuk, 2008). Consequently, there have been extensive studies on quantized CS, particularly 1-bit CS, in the past decades and a variety of algorithms have been proposed, e.g., (Zymnis et al., 2009; Dai & Milenkovic, 2011; Plan & Vershynin, 2012; 2013; Jacques et al., 2013; Xu & Kabashima, 2013; Xu et al., 2014; Awasthi et al., 2016; Meng et al., 2018; Jung et al., 2021; Liu et al., 2020; Liu & Liu, 2022). However, most existing methods are based on standard CS methods, which, therefore, inevitably inherit their inability to capture rich structures of natural signals beyond sparsity. While several recent works (Liu et al., 2020; Liu & Liu, 2022) studied 1-bit CS using generative priors, their main focuses are VAE and/or GAN, rather than SGM.

## 1.1 CONTRIBUTIONS

- We propose a novel framework, termed as Quantized Compressed Sensing with Score-based Generative Models (QCS-SGM in short), to recover unknown signals from noisy *quantized* measurements. QCS-SGM applies to an arbitrary number of quantization bits, including the extremely coarse 1-bit sign measurements. To the best of our knowledge, this is the first time that SGM has been utilized for quantized CS (QCS).

- We consider one popular SGM model called NCSNv2 (Song & Ermon, 2020) and verify the effectiveness of the proposed QCS-SGM in QCS on various real-world datasets including MNIST, Cifar-10, CelebA $64 \times 64$, and the high-resolution FFHQ $256 \times 256$. Using the pre-trained SGM as a generative prior, for both in-distribution and out-of-distribution samples, the proposed QCS-SGM significantly outperforms existing SOTA algorithms by a large margin in QCS. Perhaps surprisingly, even in the extreme case of 1-bit sign measurements, QCS-SGM can still faithfully recover the original images with much fewer measurements than the original signal dimension, as shown in Figure 1. Moreover, compared to existing methods, QCS-SGM tends to recover more natural images with good perceptual quality.

- We propose a *noise-perturbed pseudo-likelihood score* as a principled way to incorporate information from the measurements, which is one key to the success of QCS-SGM. Interestingly, when degenerated to the linear case (1), if $\mathbf{A}$ is row-orthogonal, then QCS-SGM reduces to a form similar to Jalal et al. (2021a). While the annealing term (denoted as $\gamma_t^2$ in (4) of Jalal et al. (2021a) is added heuristically as one additional hyper-parameter, it appears naturally within our framework and admits an analytical solution. More importantly, for general matrices $\mathbf{A}$, our method generalizes and significantly outperforms Jalal et al. (2021a). It is expected that the idea of noise-perturbed pseudo-likelihood score can be generalized to other conditional generative models as a general rule.

- As one sampling method, QCS-SGM can easily yield multiple samples with different random initializations, whereby one can obtain confidence intervals or uncertainty estimates.

## 1.2 RELATED WORKS

The proposed QCS-SGM is one kind of data-driven method for CS which generally include two lines of research: unsupervised and supervised. The supervised approach trains neural networks from end to end using pairs of original signals and observations (Jin et al., 2017; Zhang & Ghanem, 2018; Aggarwal et al., 2018; Wu et al., 2019; Yao et al., 2019; Gilton et al., 2019; Yang et al., 2020; Antun et al., 2020). Despite good performance on specific problems, the supervised approach is severely limited and requires re-training even with a slight change in the measurement model. The unsupervised approach, including the proposed QCS-SGM, only learns a prior and thus avoids such problem-specific training since the measurement model is only known and used for inference (Bora et al., 2017; Hand & Joshi, 2019; Asim et al., 2020; Pan et al., 2021). Among others, the CSGM framework (Jin et al., 2017) is the most popular one which learns the prior with generative models.

Recent works Liu et al. (2020); Liu & Liu (2022) extended CSGM to non-linear observations including 1-bit CS, which are the two most related studies to current work. However, the main focuses of Liu et al. (2020); Liu & Liu (2022) are limited to VAE and GAN (in particular DCGAN (Radford et al., 2015)). As is widely known, GAN suffers from unstable training and less diversity due to the adversarial training nature while VAE uses a surrogate loss. Another significant drawback of priors with VAE/GAN is that they can have large representation errors or biases due to architecture and training, which can be easily caused by inappropriate latent dimensionality and/or mode collapse (Asim et al., 2020). With the advent of SGM such as SMLD (Song & Ermon, 2019; 2020) and DDPM (Ho et al., 2020; Nichol & Dhariwal, 2021), there have been several recent studies (Jalal et al., 2021a;b; Kawar et al., 2021; 2022; Chung et al., 2022; Daras et al., 2022) in CS using SGM as a prior and outperform conventional methods. Interestingly, Daras et al. (2022) performs posterior sampling in the "latent space" of a pre-trained generative model. Nevertheless, all the existing works on SGM for CS focus on linear measurements (1) without quantization, which therefore lead to performance degradation when directly applied to the quantized case (2). Moreover, even in the linear case (1), as illustrated in Corollary 1.2 and Remark 3, the likelihood score computed in Jalal et al. (2021a;b) is not accurate for general matrices $\mathbf{A}$. By contrast, we provide a simple yet effective approximation of the likelihood score which significantly outperforms Jalal et al. (2021a) for general matrices even in the linear case. Another related work (Qiu et al., 2020) studied 1-bit CS where the sparsity is implicitly enforced via mapping a low dimensional representation through a known ReLU generative

network. Wei et al. (2019) also considers a non-linear recovery using a generative model assuming that the original signal is generated from a low-dimensional latent vector.

## 2 BACKGROUND

### 2.1 INVERSE PROBLEMS FROM QUANTIZED MEASUREMENTS

The inverse problem from noisy quantized measurements is posed as follows (Zymnis et al., 2009)

$$\mathbf{y} = \mathsf{Q}(\mathbf{A}\mathbf{x} + \mathbf{n}), \tag{2}$$

where the goal is to recover the unknown signal $\mathbf{x} \in \mathbb{R}^{N \times 1}$ from quantized measurements $\mathbf{y} \in \mathbb{R}^{M \times 1}$, where $\mathbf{A} \in \mathbb{R}^{M \times N}$ is a known linear mixing matrix, $\mathbf{n} \sim \mathcal{N}(\mathbf{n}; 0, \sigma^2 \mathbf{I})$ is an i.i.d. additive Gaussian noise with known variance, and $\mathsf{Q}(\cdot) : \mathbb{R}^{M \times 1} \to \mathcal{Q}^{M \times 1}$ is an *element-wise* [1] quantizer function which maps each element into a finite (or countable) set of codewords $\mathcal{Q}$, i.e., $y_m = \mathsf{Q}(z_m + n_m) \in \mathcal{Q}$, or equivalently $(z_m + n_m) \in \mathsf{Q}^{-1}(y_m), m = 1, 2, ..., M$, where $z_m$ is the $m$-th element of $\mathbf{z} = \mathbf{A}\mathbf{x}$.

Usually, the quantization codewords $\mathcal{Q}$ correspond to intervals (Zymnis et al., 2009), i.e., $\mathsf{Q}^{-1}(y_m) = [l_{y_m}, u_{y_m})$ where $l_{y_m}, u_{y_m}$ denote prescribed lower and upper thresholds associated with the quantized measurement $y_m$, i.e., $l_{y_m} \leq (z_m + n_m) < u_{y_m}$. For example, for a uniform quantizer with $Q$ quantization bits (resolution), the quantization codewords $\mathcal{Q} = \{q_r\}_{r=1}^{2^Q}$ consist of $2^Q$ elements, i.e.,

$$q_r = \left(2r - 2^Q - 1\right) \Delta/2, \quad r = 1, ..., 2^Q, \tag{3}$$

where $\Delta > 0$ is the quantization interval, and the lower and upper thresholds associated $q_r$ are

$$l_{q_r} = \begin{cases} -\infty, r = 1; \\ \left(r - 2^{Q-1} - 1\right)\Delta, \ r = 2, ..., 2^Q. \end{cases} \quad u_{q_r} = \begin{cases} \left(r - 2^{Q-1}\right)\Delta, \ r = 1, ..., 2^Q; \\ +\infty, \ , r = 2^Q. \end{cases} \tag{4}$$

In the extreme 1-bit case, i.e., $Q = 1$, only the sign values are observed, i.e.,

$$\mathbf{y} = \mathrm{sign}(\mathbf{A}\mathbf{x} + \mathbf{n}), \tag{5}$$

where the quantization codewords $\mathcal{Q} = \{-1, +1\}$ and the associated thresholds are $l_{-1} = -\infty, u_{-1} = 0$ and $l_{+1} = 0, u_{+1} = +\infty$, respectively.

### 2.2 SCORE-BASED GENERATIVE MODELS

For any continuously differentiable probability density function $p(\mathbf{x})$, if we have access to its score function, i.e., $\nabla_\mathbf{x} \log p(\mathbf{x})$, we can iteratively sample from it using Langevin dynamics (Turq et al., 1977; Bussi & Parrinello, 2007; Welling & Teh, 2011)

$$\mathbf{x}_t = \mathbf{x}_{t-1} + \alpha_t \nabla_{\mathbf{x}_{t-1}} \log p(\mathbf{x}_{t-1}) + \sqrt{2\alpha_t}\mathbf{z}_t, \ 1 \leq t \leq T, \tag{6}$$

where $\mathbf{z_t} \sim \mathcal{N}(\mathbf{z_t}; \mathbf{0}, \mathbf{I})$, $\alpha_t > 0$ is the step size, and $T$ is the total number of iterations. It has been demonstrated that when $\alpha_t$ is sufficiently small and $T$ is sufficiently large, the distribution of $\mathbf{x}_T$ will converge to $p(\mathbf{x})$ (Roberts & Tweedie, 1996; Welling & Teh, 2011). In practice, the score function $\nabla_\mathbf{x} \log p(\mathbf{x})$ is unknown and can be estimated using a *score network* $\mathsf{s}_{\boldsymbol{\theta}}(\mathbf{x})$ via score matching (Hyvärinen, 2006; Vincent, 2011). However, the vanilla Langevin dynamics faces a variety of challenges such as slow convergence. To address this challenge, inspired by simulated annealing (Kirkpatrick et al., 1983; Neal, 2001), Song & Ermon (2019) proposed an annealed version of Langevin dynamics, which perturbs the data with Gaussian noise of different scales and jointly estimates the score functions of noise-perturbed data distributions. Accordingly, during the inference, an annealed Langevin dynamics (ALD) is performed to leverage the information from all noise scales.

Specifically, assume that $p_\beta(\tilde{\mathbf{x}} \mid \mathbf{x}) = \mathcal{N}(\tilde{\mathbf{x}}; \mathbf{x}, \beta^2 \mathbf{I})$, and so we have $p_\beta(\tilde{\mathbf{x}}) = \int p_{\mathrm{data}}(\mathbf{x}) p_\beta(\tilde{\mathbf{x}} \mid \mathbf{x}) d\mathbf{x}$, where $p_{\mathrm{data}}(\mathbf{x})$ is the data distribution. Consider we have a sequence of noise scales $\{\beta_t\}_{t=1}^T$ satisfying $\beta_{\max} = \beta_1 > \beta_2 > \cdots > \beta_T = \beta_{\min} > 0$. The $\beta_{\min} \to 0$ is small enough so that $p_{\beta_{\min}}(\tilde{\mathbf{x}}) \approx p_{\mathrm{data}}(\mathbf{x})$, and $\beta_{\max}$ is large enough so that $p_{\beta_{\max}}(\tilde{\mathbf{x}}) \approx \mathcal{N}(\mathbf{x}; 0, \beta_{\max}^2 I)$. The noise

---

[1] While more sophisticated quantization schemes exist (Dirksen, 2019), here we focus on memoryless scalar quantization due to its popularity and simplicity where the quantizer acts element-wise.

conditional score network (NCSN) $s_{\boldsymbol{\theta}}(\mathbf{x}, \beta)$ proposed in Song & Ermon (2019) aims to estimate the score function of each $p_{\beta_t}(\tilde{\mathbf{x}})$ by optimizing the following weighted sum of score matching objective

$$\boldsymbol{\theta}^* = \arg\min_{\boldsymbol{\theta}} \sum_{t=1}^{T} \mathbb{E}_{p_{\text{data}}(\mathbf{x})} \mathbb{E}_{p_{\beta_t}(\tilde{\mathbf{x}}|\mathbf{x})} \left[ \|s_{\boldsymbol{\theta}}(\tilde{\mathbf{x}}, \beta_t) - \nabla_{\tilde{\mathbf{x}}} \log p_{\beta_t}(\tilde{\mathbf{x}} \mid \mathbf{x})\|_2^2 \right]. \tag{7}$$

After training the NCSN, for each noise scale, we can run $K$ steps of Langevin MCMC to obtain a sample for each $p_{\beta_t}(\tilde{\mathbf{x}})$ as

$$\mathbf{x}_t^k = \mathbf{x}_t^{k-1} + \alpha_t s_{\boldsymbol{\theta}}(\mathbf{x}_t^{k-1}, \beta_t) + \sqrt{2\alpha_t}\mathbf{z}_t^k, \ k = 1, 2, ..., K. \tag{8}$$

The sampling process is repeated for $t = 1, 2, ..., T$ sequentially with $\mathbf{x}_1^0 \sim \mathcal{N}(\mathbf{x}; \mathbf{0}, \beta_{\max}^2 \mathbf{I})$ and $\mathbf{x}_{t+1}^0 = \mathbf{x}_t^K$ when $t < T$. As shown in Song & Ermon (2019), when $K \to \infty$ and $\alpha_t \to 0$ for all $t$, the final sample $\mathbf{x}_T^K$ will become an exact sample from $p_{\beta_{\min}}(\tilde{\mathbf{x}}) \approx p_{\text{data}}(\mathbf{x})$ under some regularity conditions. Later, by a theoretical analysis of the learning and sampling process of NCSN, an improved version of NCSN, termed NCSNv2, was proposed in Song & Ermon (2020) which is more stable and can scale to various datasets with high resolutions.

## 3 QUANTIZED CS WITH SGM

In the case of quantized measurements (2) with access to $\mathbf{y}$, the goal is to sample from the posterior distribution $p(\mathbf{x} \mid \mathbf{y})$ rather than $p(\mathbf{x})$. Subsequently, the Langevin dynamics in (6) becomes

$$\mathbf{x}_t = \mathbf{x}_{t-1} + \alpha_t \nabla_{\mathbf{x}_{t-1}} \log p(\mathbf{x}_{t-1} \mid \mathbf{y}) + \sqrt{2\alpha_t}\mathbf{z}_t, \ 1 \le t \le T, \tag{9}$$

where the conditional (*posterior*) score $\nabla_{\mathbf{x}} \log p(\mathbf{x} \mid \mathbf{y})$ is required. One direct solution is to specially train a score network work that matches the score $\nabla_{\mathbf{x}} \log p(\mathbf{x} \mid \mathbf{y})$. However, this method is rather inflexible and needs to re-train the network when confronted with different measurements. Instead, inspired by conditional diffusion models (Jalal et al., 2021a), we resort to computing the posterior score using the Bayesian rule from which $\nabla_{\mathbf{x}} \log p(\mathbf{x} \mid \mathbf{y})$ is decomposed into two terms

$$\nabla_{\mathbf{x}} \log p(\mathbf{x} \mid \mathbf{y}) = \nabla_{\mathbf{x}} \log p(\mathbf{x}) + \nabla_{\mathbf{x}} \log p(\mathbf{y} \mid \mathbf{x}), \tag{10}$$

which include the unconditional score $\nabla_{\mathbf{x}} \log p(\mathbf{x})$ (we call it *prior score*), and the conditional score $\nabla_{\mathbf{x}} \log p(\mathbf{y} \mid \mathbf{x})$ (we call it *likelihood score*), respectively. The prior score $\nabla_{\mathbf{x}} \log p(\mathbf{x})$ can be easily obtained using a trained score network such as NCSN or NCSNv2, so the remaining goal is to compute the likelihood score $\nabla_{\mathbf{x}} \log p(\mathbf{y} \mid \mathbf{x})$ from the quantized measurements.

### 3.1 NOISE-PERTURBED PSEUDO-LIKELIHOOD SCORE

In the case without noise perturbing in $\mathbf{x}$, one can calculate the likelihood score $\nabla_{\mathbf{x}} \log p(\mathbf{y} \mid \mathbf{x})$ directly from the measurement process (2) and then substitute it into (10) to obtain the posterior score $\nabla_{\mathbf{x}} \log p(\mathbf{x} \mid \mathbf{y})$. However, this does not work well and might even diverge (Jalal et al., 2021a). The reason is that in NCSN/NCSNv2, as shown in (7), the estimated score is not the original prior score $\nabla_{\mathbf{x}} \log p_{\text{data}}(\mathbf{x})$ but rather a noise-perturbed prior score $\nabla_{\tilde{\mathbf{x}}} \log p_{\beta_t}(\tilde{\mathbf{x}})$ for each noise scale $\beta_t$. Therefore, the likelihood score directly computed from (2) does not match the noise-perturbed prior score. To address this problem, we propose to calculate the *noise-perturbed pseudo-likelihood score* $\nabla_{\tilde{\mathbf{x}}} \log p_{\beta_t}(\mathbf{y} \mid \tilde{\mathbf{x}})$ by explicitly accounting for such the noise perturbation in NCSN/NCSNv2. Unfortunately, for NCSN/NCSNv2, the $\nabla_{\tilde{\mathbf{x}}} \log p_{\beta_t}(\mathbf{y} \mid \tilde{\mathbf{x}})$ is intractable in the general case. To this end, we propose a simple yet effective approximation by introducing two assumptions.

**Assumption 1.** *The prior $p(\mathbf{x})$ is uninformative w.r.t. $p(\tilde{\mathbf{x}} \mid \mathbf{x})$, i.e., $p(\mathbf{x} \mid \tilde{\mathbf{x}}) \propto p(\tilde{\mathbf{x}} \mid \mathbf{x})$.*

**Remark 1.** *This assumption is asymptotically accurate when the perturbed noise in $\tilde{\mathbf{x}}$ becomes negligible, i.e., $\beta_t \to 0$ when $t$ is large. An illustration of this is shown in Appendix A.*

**Assumption 2.** *The matrix $\mathbf{A}$ is row-orthogonal, i.e., $\mathbf{A}\mathbf{A}^T$ is a diagonal matrix.*

**Remark 2.** *The measurement matrices $\mathbf{A}$ are usually carefully designed in CS. Interestingly, most popular CS matrices $\mathbf{A}$ satisfy this assumption, e.g., discrete Fourier transform (DFT) in MRI, discrete cosine transform (DCT), Hadamard matrices, as well as general row-orthogonal matrices Vehkaperä et al. (2016). For another class of widely used i.i.d. Gaussian matrices, this assumption is approximated satisfied, as verified in Appendix B. Notably, it has been theoretically demonstrated*

*that row-orthogonal matrices are mostly preferred for CS due to both superior performance and easy implementation (Vehkaperä et al., 2016). Therefore, this assumption does not impose a serious limitation in the case of CS, and we leave the study of general $\mathbf{A}$ as a future work.*

Subsequently, we obtain our main result as shown in Theorem 1.

**Theorem 1.** *(noise-perturbed pseudo-likelihood score) For each noise scale $\beta_t > 0$, if $p_{\beta_t}(\tilde{\mathbf{x}} \mid \mathbf{x}) = \mathcal{N}(\tilde{\mathbf{x}}; \mathbf{x}, \beta_t^2 \mathbf{I})$, under Assumption 1 and Assumption 2, the noise-perturbed pseudo-likelihood score $\nabla_{\tilde{\mathbf{x}}} \log p_{\beta_t}(\mathbf{y} \mid \tilde{\mathbf{x}})$ for the quantized measurements $\mathbf{y}$ in (2) can be computed as*

$$\nabla_{\tilde{\mathbf{x}}} \log p_{\beta_t}(\mathbf{y} \mid \tilde{\mathbf{x}}) = \mathbf{A}^T \mathbf{G}(\beta_t, \mathbf{y}, \mathbf{A}, \tilde{\mathbf{x}}), \tag{11}$$

*where $\mathbf{G}(\beta_t, \mathbf{y}, \mathbf{A}, \tilde{\mathbf{x}}) = [g_1, g_2, ..., g_M]^T \in \mathbb{R}^{M \times 1}$ with each element being*

$$g_m = \frac{\exp\left(-\frac{\tilde{u}_{y_m}^2}{2}\right) - \exp\left(-\frac{\tilde{l}_{y_m}^2}{2}\right)}{\sqrt{\sigma^2 + \beta_t^2 \|\mathbf{a}_m^T\|_2^2} \int_{\tilde{l}_{y_m}}^{\tilde{u}_{y_m}} \exp\left(-\frac{t^2}{2}\right) dt}, \quad m = 1, 2, ..., M, \tag{12}$$

*where $\mathbf{a}_m^T \in \mathbb{R}^{1 \times N}$ denotes the $m$-th row vector of $\mathbf{A}$ and*

$$\tilde{u}_{y_m} = \frac{\mathbf{a}_m^T \tilde{\mathbf{x}} - u_{y_m}}{\sqrt{\sigma^2 + \beta_t^2 \|\mathbf{a}_m^T\|_2^2}}, \quad \tilde{l}_{y_m} = \frac{\mathbf{a}_m^T \tilde{\mathbf{x}} - l_{y_m}}{\sqrt{\sigma^2 + \beta_t^2 \|\mathbf{a}_m^T\|_2^2}}, \tag{13}$$

*Proof.* The proof is shown in Appendix C. □

In the extreme 1-bit case, a more concise result can be obtained, as shown in Corollary 1.1:

**Corollary 1.1.** *(noise-perturbed pseudo-likelihood score in the 1-bit case) For each noise scale $\beta_t > 0$, if $p_{\beta_t}(\tilde{\mathbf{x}} \mid \mathbf{x}) = \mathcal{N}(\tilde{\mathbf{x}}; \mathbf{x}, \beta_t^2 \mathbf{I})$, under Assumption 1 and Assumption 2, the noise-perturbed pseudo-likelihood score $\nabla_{\tilde{\mathbf{x}}} \log p_{\beta_t}(\mathbf{y} \mid \tilde{\mathbf{x}})$ for the sign measurements $\mathbf{y} = \text{sign}(\mathbf{A}\mathbf{x} + \mathbf{n})$ in (5) can be computed as*

$$\nabla_{\tilde{\mathbf{x}}} \log p_{\beta_t}(\mathbf{y} \mid \tilde{\mathbf{x}}) = \mathbf{A}^T \mathbf{G}(\beta_t, \mathbf{y}, \mathbf{A}, \tilde{\mathbf{x}}), \tag{14}$$

*where $\mathbf{G}(\beta_t, \mathbf{y}, \mathbf{A}, \tilde{\mathbf{x}}) = [g_1, g_2, ..., g_M]^T \in \mathbb{R}^{M \times 1}$ with each element being*

$$g_m = \left[\frac{1 + y_m}{2\Phi(\tilde{z}_m)} - \frac{1 - y_m}{2(1 - \Phi(\tilde{z}_m))}\right] \frac{\exp\left(-\frac{\tilde{z}_m^2}{2}\right)}{\sqrt{2\pi(\sigma^2 + \beta_t^2 \|\mathbf{a}_m^T\|_2^2)}}, \quad m = 1, 2, ..., M, \tag{15}$$

*where $\tilde{z}_m = \frac{\mathbf{a}_m^T \tilde{\mathbf{x}}}{\sqrt{\sigma^2 + \beta_t^2 \|\mathbf{a}_m^T\|_2^2}}$ and $\Phi(z) = \frac{1}{\sqrt{2\pi}} \int_{-\infty}^{z} e^{-\frac{t^2}{2}} dt$ is the cumulative distribution function (CDF) of the standard normal distribution.*

*Proof.* It can be easily proved following the proof in Theorem 1. □

Interestingly, if no quantization is used, or equivalently $Q \to \infty$, the model in (2) reduces to the standard linear model (1). Accordingly, results in Theorem 1 can be modified as in Corollary 1.2:

**Corollary 1.2.** *(noise-perturbed pseudo-likelihood score without quantization) For each noise scale $\beta_t > 0$, if $p_{\beta_t}(\tilde{\mathbf{x}} \mid \mathbf{x}) = \mathcal{N}(\tilde{\mathbf{x}}; \mathbf{x}, \beta_t^2 \mathbf{I})$, under Assumption 1, the noise-perturbed pseudo-likelihood score $\nabla_{\tilde{\mathbf{x}}} \log p_{\beta_t}(\mathbf{y} \mid \tilde{\mathbf{x}})$ for linear measurements $\mathbf{y} = \mathbf{A}\mathbf{x} + \mathbf{n}$ in (1) can be computed as*

$$\nabla_{\tilde{\mathbf{x}}} \log p_{\beta_t}(\mathbf{y} \mid \tilde{\mathbf{x}}) = \mathbf{A}^T \left(\sigma^2 \mathbf{I} + \beta_t^2 \mathbf{A}\mathbf{A}^T\right)^{-1} \left(\mathbf{y} - \mathbf{A}\tilde{\mathbf{x}}\right). \tag{16}$$

*Moreover, under Assumption 2, the $\nabla_{\tilde{\mathbf{x}}} \log p_{\beta_t}(\mathbf{y} \mid \tilde{\mathbf{x}})$ can be further simplified as*

$$\nabla_{\tilde{\mathbf{x}}} \log p_{\beta_t}(\mathbf{y} \mid \tilde{\mathbf{x}}) = \mathbf{A}^T \mathbf{G}_{\text{linear}}(\beta_t, \mathbf{y}, \mathbf{A}, \tilde{\mathbf{x}}), \tag{17}$$

*where $\mathbf{G}_{\text{linear}}(\beta_t, \mathbf{y}, \mathbf{A}, \tilde{\mathbf{x}}) = [g_1, g_2, ..., g_M]^T \in \mathbb{R}^{M \times 1}$ with each element being*

$$g_m = \frac{y_m - \mathbf{a}_m^T \tilde{\mathbf{x}}}{\sigma^2 + \beta_t^2 \|\mathbf{a}_m^T\|_2^2}, \quad m = 1, 2, ..., M. \tag{18}$$

*Proof.* The proof is shown in Appendix D. □

**Remark 3.** *In contrast to the quantized case, here we obtain a closed-form result (16) for any kind of matrices* $\mathbf{A}$. *Interestingly, if Assumption 2 is further imposed, it reduces to (18), which is similar to Jalal et al. (2021a) where the annealing term* $\beta_t^2 \left\| \mathbf{a}_m^T \right\|_2^2$ *in (18) corresponds to* $\gamma_t^2$ *in Jalal et al. (2021a). However,* $\gamma_t^2$ *in Jalal et al. (2021a) is added heuristically as an additional hyper-parameter while we derive it in closed-form under the principled perturbed pseudo-likelihood score. More importantly, for general matrices* $\mathbf{A}$, *our closed-form result (16) significantly outperforms Jalal et al. (2021a), as demonstrated in Appendix E. Therefore, we not only explain the necessity of an annealing term in Jalal et al. (2021a) but also extend and improve Jalal et al. (2021a) in the general case.*

### 3.2 POSTERIOR SAMPLING VIA ANNEALED LANGEVIN DYNAMICS

By combining the prior score from SGM and noise-perturbed pseudo-likelihood score in Theorem 1, we obtain the resultant Algorithm 1, namely, Quantized CS with SGM (QCS-SGM in short).

---

**Algorithm 1:** Quantized Compressed Sensing with SGM (QCS-SGM)

---

**Input:** $\{\beta_t\}_{t=1}^T, \epsilon, K, \mathbf{y}, \mathbf{A}, \sigma^2$, quantization codewords $\mathcal{Q}$ and thresholds $\{[l_q, u_q] | q \in \mathcal{Q}\}$
**Initialization:** $\mathbf{x}_1^0 \sim \mathcal{U}(0, 1)$

1   **for** $t = 1$ **to** $T$ **do**
2      $\alpha_t \leftarrow \epsilon \beta_t^2 / \beta_T^2$
3      **for** $k = 1$ **to** $K$ **do**
4          Draw $\mathbf{z}_t^k \sim \mathcal{N}(\mathbf{0}, \mathbf{I})$
5          Compute $\mathbf{G}(\beta_t, \mathbf{y}, \mathbf{A}, \mathbf{x}_t^{k-1})$ as (12) (or (15) for 1-bit)
6          $\mathbf{x}_t^k = \mathbf{x}_t^{k-1} + \alpha_t \left[ s_{\boldsymbol{\theta}}(\mathbf{x}_t^{k-1}, \beta_t) + \mathbf{A}^T \mathbf{G}(\beta_t, \mathbf{y}, \mathbf{A}, \mathbf{x}_t^{k-1}) \right] + \sqrt{2\alpha_t} \mathbf{z}_t^k$
7      $\mathbf{x}_{t+1}^0 \leftarrow \mathbf{x}_t^K$

**Output:** $\hat{\mathbf{x}} = \mathbf{x}_T^K$

---

## 4 EXPERIMENTS

We empirically demonstrate the efficacy of the proposed QCS-SGM in various scenarios. The widely-used i.i.d. Gaussian matrix $\mathbf{A}$, i.e., $A_{ij} \sim \mathcal{N}(0, 1/M)$ are considered. *The code is available at https://github.com/mengxiangming/QCS-SGM.*

**Datasets**: Three popular datasets are considered: MNIST (LeCun & Cortes, 2010) , Cifar-10 (Krizhevsky & Hinton, 2009), and CelebA (Liu et al., 2015), and the high-resolution Flickr Faces High Quality (FFHQ) (Karras et al., 2018). MNIST (LeCun & Cortes, 2010) are grayscale images of size $28 \times 28$ pixels so that the input dimension for MNIST is $N = 28 \times 28 = 784$ per image. Cifar-10 (Krizhevsky & Hinton, 2009) consists of natural RGB images of size $32 \times 32$ pixels, resulting in $N = 32 \times 32 \times 3 = 3072$ inputs per image. For CelebA dataset (Liu et al., 2015), we cropped each face image to a $64 \times 64$ RGB image, resulting in $N = 64 \times 64 \times 3 = 12288$ inputs per image. The FFHQ are high-resolution RGB images of size $256 \times 256$, so that $N = 256 \times 256 \times 3 = 196608$ per image. Note that to evaluate the out-of-distribution (OOD) performance, we also cropped FFHQ to size $64 \times 64$, as OOD samples for CelebA. All images are normalized to range $[0, 1]$.

**QCS-SGM**: Regarding the SGM models, we adopt the popular NCSNv2 (Song & Ermon, 2020) in all cases. Specifically, for MNIST, we train a NCSNv2 (Song & Ermon, 2020) model on the MNIST training dataset with a similar training set up as Cifar10 in Song & Ermon (2020), while we directly use the for Cifar-10, and CelebA, and FFHQ, which are available in this Link. Therefore, the prior score can be estimated using these pre-trained NCSNv2 models. After observing the quantized measurements as (2), we can infer the original $\mathbf{x}$ by posterior sampling via QCS-SGM in Algorithm 1. It is important to note that, *we select images* $\mathbf{x}$ *that are unseen by the pre-traineirec SGM models*. For details of the experimental setting, please refer to Appendix F. We have submitted the code in the supplementary material and will release it upon acceptance.

### 4.1 1-BIT QUANTIZATION

First, we perform experiments on the extreme case, i.e., 1-bit measurements. Specifically, we consider images of the MNIST (LeCun & Cortes, 2010) and CelebA datasets (Liu et al., 2015) in the same setting as Liu & Liu (2022). Apart from QCS-SGM, we also show results of LASSO (Tibshirani, 1996) (for CelebA, Lasso with a DCT basis (Ahmed et al., 1974) is used), CSGM (Bora et al., 2017),

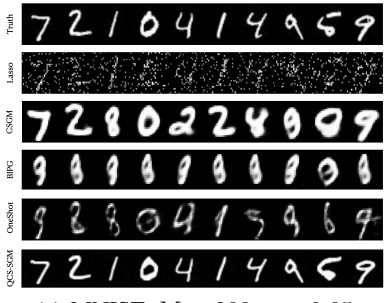 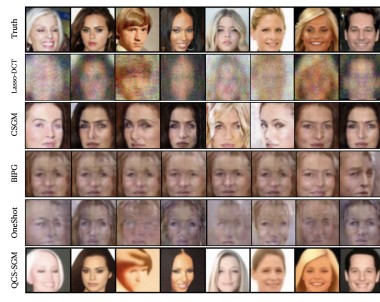

(a) MNIST, $M = 200, \sigma = 0.05$          (b) CelebA, $M = 4000, \sigma = 0.001$

Figure 2: Typical reconstructed images from 1-bit measurements on MNIST and CelebA. QCS-SGM faithfully recovers original images from 1-bit measurements even when $M \ll N$. Compared to other methods, QCS-SGM recovers more natural images with good perceptual quality.

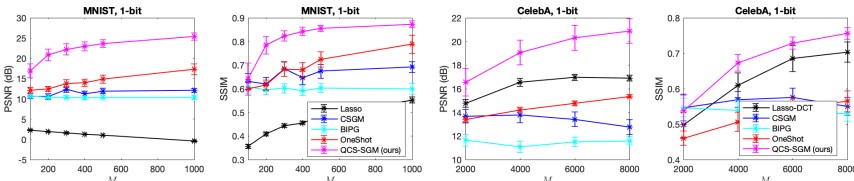

Figure 3: Quantitative comparisons based on different metrics for 1-bit MNIST and CelebA. QCS-SGM remarkably outperforms all other methods under all metrics.

BIPG (Liu et al., 2020), and OneShot (Liu & Liu, 2022). For Lasso (Lasso-DCT), CSGM, BIPG, and OneShot, we follow the default setting as the open-sourced code of Liu & Liu (2022).

The typical reconstructed images from 1-bit measurements with fixed $M \ll N$ are shown in Figure 2 for both MNIST and CelebA. Perhaps surprisingly, QCS-SGM can faithfully recover the images from 1-bit measurements with $M \ll N$ measurements while other methods fail or only recover quite vague images. To quantitatively evaluate the effect of $M$, we compare different algorithms using three popular metrics, i.e., peak signal-to-noise ratio (PSNR) and structural similarity index (SSIM) for different $M$, and the results are shown in Figure 3. It can be seen that, under both metrics, the proposed QCS-SGM significantly outperforms all the other algorithms by a large margin. An investigation of the effect of (pre-quantization) Gaussian noise is shown in Figure 23 in Appendix J. Please refer to Appendix H for more results.

**Multiple samples** & **Uncertainty Estimates**: It is worth pointing out that, as one kind of sampling method, QCS-SGM can yield multiple samples with different random initialization so that we can easily obtain confidence intervals or uncertainty estimates of the reconstructed results. Please refer to Appendix G for more typical samples of QCS-SGM. In contrast, OneShot (Liu & Liu, 2022) can easily get stuck in a local minimum with different random initializations (Liu & Liu, 2022).

### 4.2 MULTI-BIT QUANTIZATION

Then, we evaluate the efficacy of QCS-SGM in the case of multi-bit quantization, e.g., 2-bit, 3-bit. The results on Cifar-10 and CelebA are shown in Figure 4. The results of the linear case without quantization are also shown for comparison using (18) in Corollary 1.2. As expected, with the increase of quantization resolution, the reconstruction performances get better. One typical result for high-resolution FFHQ $256 \times 256$ images is shown in Figure 1. Moreover, to evaluate the quantization effect under "fixed budget" setting, i.e., $Q \times M$ remain the same for different $Q$ and $M$, we conduct experiments for CelebA in the fixed case of $Q \times M = 12288$. Interestingly, as shown in Table 1, under 'fixed budget", while there is an apparent increase in PSNR for the multi-bit case, the perception metric SSIM remains about the same as 1-bit case. For more results, please refer to Appendix H.

### 4.3 OUT-OF-DISTRIBUTION PERFORMANCE

In practice, the signals of interest may be out-of-distribution (OOD) because the available training dataset has bias and is unrepresentative of the true underlying distribution. To assess the performance

| CelebA ($64 \times 64$) | 1-bit, $M = 12288$ | | 2-bit, $M = 6144$ | | 3-bit, $M = 4096$ | |
|---|---|---|---|---|---|---|
| **Method** | PSNR ↑ | SSIM ↑ | PSNR ↑ | SSIM ↑ | PSNR ↑ | SSIM ↑ |
| QCS-SGM (ours) | **20.1** | **0.81** | **24.3** | **0.78** | **25.8** | **0.80** |
| Lasso-DCT | 16.2 | 0.68 | 16.9 | 0.69 | 18.2 | 0.74 |
| CSGM | 17.6 | 0.68 | 18.6 | 0.70 | 18.9 | 0.72 |
| BIPG | 11.6 | 0.57 | 11.59 | 0.47 | 11.45 | 0.53 |
| OneShot | 16.1 | 0.62 | 15.3 | 0.55 | 14.75 | 0.54 |

Table 1: Quantitative comparison of different methods for $Q$-bit CS with "fixed budget", i.e., $Q \times M$ is fixed. Results are averaged over a validation set of size 100. Reconstructed images are shown in Appendix H.

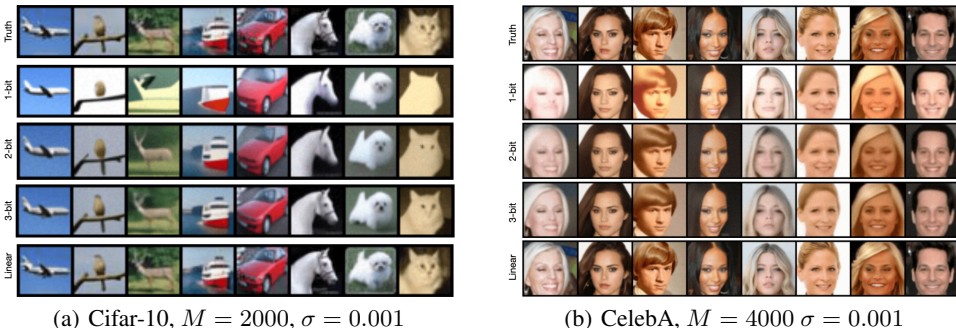

(a) Cifar-10, $M = 2000$, $\sigma = 0.001$          (b) CelebA, $M = 4000$ $\sigma = 0.001$

Figure 4: Results of QCS-SGM for Cifar-10 and CelebA images under different quantization bits.

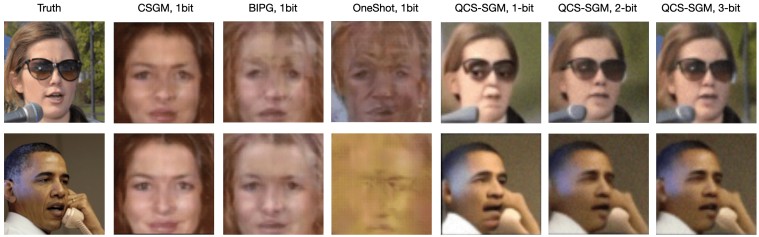

Figure 5: Results on out-of-distribution (OOD) FFHQ dataset from 1-bit (2-bit and 3-bit are also shown for QCS-SGM) measurements. $M = 10000, \sigma = 0.001$.

of QCS-SGM for OOD datasets, we choose the CelebA as the training set while evaluating results on OOD images randomly sampled from the FFHQ dataset, which have features of variation that are rare among CelebA images (e.g., skin tone, age, beards, and glasses) (Jalal et al., 2021b). As shown in Figure 5, even on OOD images, the proposed QCS-SGM can still obtain significantly competitive results compared with other existing algorithms.

## 5    CONCLUSION

In this paper, a novel framework called Quantized Compressed Sensing with Score-based Generative Models (QCS-SGM) is proposed to reconstruct a high-dimensional signal from noisy quantized measurements. Thanks to the power of SGM in capturing the rich structure of natural signals beyond simple sparsity, the proposed QCS-SGM significantly outperforms existing SOTA algorithms by a large margin under coarsely quantized measurements on a variety of experiments, for both in-distribution and out-of-distribution datasets. There are still some limitations in QCS-SGM. For example, it is limited to the case where measurement matrix $\mathbf{A}$ is (approximately) row-orthogonal. The generalization QCS-SGM to general matrices is an important future work. In addition, at present the training of SGM models is still very expensive, which hinders the applicability of QCS-SGM in the resource-limited scenario when no pre-trained models are available. Compared to traditional CS algorithms without data-driven prior, the possible lack of enough training data also imposes a great challenge for QCS-SGM. It is therefore important to address these limitations in the future. Nevertheless, as it is a research topic with rapid progress, we believe that SGM could open up new opportunities for quantized CS, and sincerely hope that our work as a first step could inspire more studies on this fascinating topic.

ACKNOWLEDGEMENTS

X. Meng would like to thank Jiulong Liu for his help in understanding their code of OneShot. This work was supported by JSPS KAKENHI Nos. 17H00764, and 19H01812, 22H05117, and JST CREST Grant Number JPMJCR1912, Japan.

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

## A    VERIFICATION OF ASSUMPTION 1

Mathematically, we can write the exact $p(\mathbf{y} \mid \tilde{\mathbf{x}})$ as

$$p(\mathbf{y} \mid \tilde{\mathbf{x}}) = \int p(\mathbf{y} \mid \mathbf{x}) p(\mathbf{x} \mid \tilde{\mathbf{x}}) d\mathbf{x}, \tag{19}$$

where from the Bayes' rule,

$$p(\mathbf{x} \mid \tilde{\mathbf{x}}) = \frac{p(\tilde{\mathbf{x}} \mid \mathbf{x}) p(\mathbf{x})}{\int p(\tilde{\mathbf{x}} \mid \mathbf{x}) p(\mathbf{x}) d\mathbf{x}}. \tag{20}$$

For NCSN/NCSNv2, recall that the likelihood $p(\tilde{\mathbf{x}} \mid \mathbf{x})$ follows a Gaussian distribution $\mathcal{N}(\tilde{\mathbf{x}}; \mathbf{x}, \beta^2 \mathbf{I})$, where $\beta$ is the variance of the perturbed Gaussian noise and, by definition, $\beta \to 0$ in later steps of the reverse diffusion process in NCSN/NCSNv2. As a result, from (20), the likelihood $p(\tilde{\mathbf{x}} \mid \mathbf{x})$ will dominate the posterior $p(\mathbf{x} \mid \tilde{\mathbf{x}})$ as $\beta \to 0$ and therefore $p(\mathbf{x} \mid \tilde{\mathbf{x}}) \propto p(\tilde{\mathbf{x}} \mid \mathbf{x})$, indicating that the prior $p(\mathbf{x})$ becomes uninformative. Note that this is particularly the case when the entropy of $p(\mathbf{x})$ is high, which is usually the case for generative models capable of generating diverse images. While this assumption is crude at the beginning of the reverse diffusion process when the noise variance $\beta$ is large, it is asymptotically accurate as the process goes forward. The effectiveness of this assumption is also empirically supported by various experiments in Section 4.

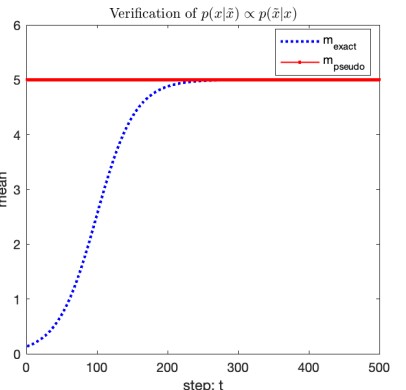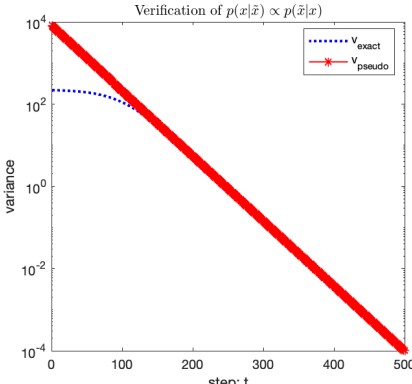

Figure 6: Comparison of the exact mean and variance of $p(x \mid \tilde{x})$ with the pseudo mean and variance under the uninformative assumption, i.e., $p(x \mid \tilde{x}) \propto p(\tilde{x} \mid x)$ in the toy scalar Gaussian example. In this plot, we set $\beta_{\max} = 90$, $\beta_{\min} = 0.01$, $T = 500$, which are the same as the setting of NCSNv2 for CelebA. The $\tilde{x}$ and the prior standard deviation $\sigma_0$ is set to be $\sigma_0 = 15$. It can be seen that the approximated values approach the exact values very quickly, verifying the effectiveness of the Assumption 1 for this toy example.

**A toy example:** In the following, we illustrate the assumption in a toy example where $\mathbf{x}$ reduces to a scalar random variable $x$ and the associated prior $p(x)$ follows a Gaussian distribution, i.e., $p(x) = \mathcal{N}(x; 0, \sigma_0^2)$, where $\sigma^2$ is the prior variance. The likelihood $p(\tilde{x} \mid \mathbf{x})$ in this case is simply $p(\tilde{x} \mid x) = \mathcal{N}(\tilde{x}; x, \beta^2)$. Then, from (20), after some algebra, it can be computed that the posterior distribution $p(x \mid \tilde{x})$ is

$$p(x \mid \tilde{x}) = \mathcal{N}(\tilde{x}; m_{\text{exact}}, v_{\text{exact}}) \tag{21}$$

where

$$m_{\text{exact}} = \frac{\sigma_0^2}{\sigma_0^2 + \beta^2} \tilde{x}, \ v_{\text{exact}} = \frac{\sigma_0^2 \beta^2}{\sigma_0^2 + \beta^2}. \tag{22}$$

Under the Assumption 1, i.e., $p(x \mid \tilde{x}) \propto p(\tilde{x} \mid x)$, we obtain an approximation of $p(x \mid \tilde{x})$ as follows

$$p(x \mid \tilde{x}) \simeq \tilde{p}(x \mid \tilde{x}) = \mathcal{N}(x; m_{\text{pseudo}}, v_{\text{pseudo}}), \tag{23}$$

where

$$m_{\text{pseudo}} = \tilde{x}, \ v_{\text{pseudo}} = \beta^2. \tag{24}$$

By comparing the exact result (22) and approximation result (24), it can be easily seen that for a fixed $\sigma_0^2 > 0$, as $\beta \to 0$, we have $m_{\text{pseudo}} \to m_{\text{post}}$ and $v_{\text{pseudo}} \to v_{\text{post}}$. In Figure 6, similar to NCSN/NCSNv2, we anneal $\beta$ as $\beta_t = \beta_{\max}(\frac{\beta_{\min}}{\beta_{\max}})^{\frac{t-1}{T-1}}$ geometrically and compare $m_{\text{pseudo}}, v_{\text{pseudo}}$ with $m_{\text{exact}}, v_{\text{exact}}$ as $t$ increase from 1 to $T$. It can be seen in Figure 6 that the approximated values $m_{\text{pseudo}}, v_{\text{pseudo}}$, especially the variance $v_{\text{pseudo}}$, approach to the exact values $m_{\text{exact}}, v_{\text{exact}}$ very quickly, verifying the effectiveness of the Assumption 1 under this toy example.

## B  VERIFICATION OF ASSUMPTION 2

Before verifying it in the i.i.d. Gaussian case, we first briefly add a comment on the general case when $\mathbf{A}\mathbf{A}^T$ is not an exact diagonal matrix. As demonstrated later in Appendix C, the Assumption 2 is introduced to ensure that the covariance matrix $\sigma^2\mathbf{I} + \beta_t^2\mathbf{A}\mathbf{A}^T$ of Gaussian noise is diagonal so that we can obtain a closed-form solution for the likelihood score in the quantized case, which is otherwise intractable. If

Suppose that the elements of $\mathbf{A}$ follow i.i.d. Gaussian, i.e., $A_{ij} \sim \mathcal{N}(0, \sigma^2)$. Next, we investigate the elements of the matrix $\mathbf{C} = \mathbf{A}\mathbf{A}^T = \{C_{ij}\}, i, j = 1...M$.

Regarding the diagonal elements $C_{ii}$, by definition, it reads as

$$C_{ii} = \sum_{n=1...N} A_{in}^2, \ i = 1...M. \tag{25}$$

As $C_{ii}$ is the sum of square of $N$ i.i.d. Gaussian random variables $A_{ij}, \ j = 1...N$, $C_{ii}$ follows a Gamma distribution, i.e., $\Gamma(\frac{N}{2}, 2\sigma^2)$. The mean and variance of $C_{ii}$ can be computed as $N\sigma^2$ and $2N\sigma^4$.

Regarding the off-diagonal elements $C_{ij}, i \neq j$, by definition, it reads as

$$C_{ij} = \sum_{n=1...N} A_{in}A_{jn}, \ i, j = 1...M, i \neq j. \tag{26}$$

As $A_{in}$ and $A_{jn}$ are independent Gaussian for $i \neq j$, it can be computed that the mean and variance of $C_{ij}$ are 0 and $N\sigma^4$, respectively. When $\sigma^2 = 1/M$ and $M = \alpha N$, where $\alpha > 0$ is the constant measurement ratio, the variance of $C_{ij}$ is $\frac{1}{\alpha^2 N} \to 0$ as $N \to \infty$. As a result, all the off-diagonal elements of $\mathbf{A}\mathbf{A}^T$ will tend to zero as $N \to \infty$. From another perspective, in Figure 7, we compute the ratio between the average magnitude of off-diagonal elements $C_{ij}$ and the diagonal elements $C_{ii}$ when $M = \alpha N$ with $\alpha = 0.5$. It can be seen that as $N$ increases, the magnitude of the off-diagonal elements becomes negligible. As a result, when $\mathbf{A}$ is i.i.d. Gaussian, the matrix $\mathbf{A}\mathbf{A}^T$ can be well approximated as a diagonal matrix in the high-dimensional case.

## C  PROOF OF THEOREM 1

*Proof.* Let us denote $\mathbf{z} = \mathbf{A}\mathbf{x}$. For each noise scale $\beta_t > 0$, under the Assumption 1, we obtain

$$p_{\beta_t}(\mathbf{x} \mid \tilde{\mathbf{x}}) \propto p_{\beta_t}(\tilde{\mathbf{x}} \mid \mathbf{x})$$
$$\sim \mathcal{N}(\mathbf{x}; \tilde{\mathbf{x}}, \beta_t^2\mathbf{I}) \tag{27}$$

then we can equivalently write

$$\mathbf{x} = \tilde{\mathbf{x}} + \beta_t\mathbf{w}, \tag{28}$$

where $\mathbf{w} \sim \mathcal{N}(\mathbf{0}, \mathbf{I})$. As a result, $\mathbf{z} = \mathbf{A}\mathbf{x} = \mathbf{A}(\tilde{\mathbf{x}} + \beta_t\mathbf{w}) = \mathbf{A}\tilde{\mathbf{x}} + \beta_t\mathbf{A}\mathbf{w}$. Then, from (2), we obtain

$$\mathbf{y} = \mathsf{Q}(\mathbf{A}\tilde{\mathbf{x}} + \tilde{\mathbf{n}}), \tag{29}$$

where $\tilde{\mathbf{n}} = \mathbf{n} + \beta_t\mathbf{A}\mathbf{w}$. Since $\mathbf{n} \sim \mathcal{N}(\mathbf{0}, \sigma^2\mathbf{I})$ and $\mathbf{w} \sim \mathcal{N}(\mathbf{0}, \mathbf{I})$ and are independent to each other, it can be concluded that $\tilde{\mathbf{n}}$ is also Gaussian with mean zero and covariance $\sigma^2\mathbf{I} + \beta_t^2\mathbf{A}\mathbf{A}^T$, i.e., $\tilde{\mathbf{n}} \sim \mathcal{N}(\tilde{\mathbf{n}}; \mathbf{0}, \sigma^2\mathbf{I} + \beta_t^2\mathbf{A}\mathbf{A}^T)$.

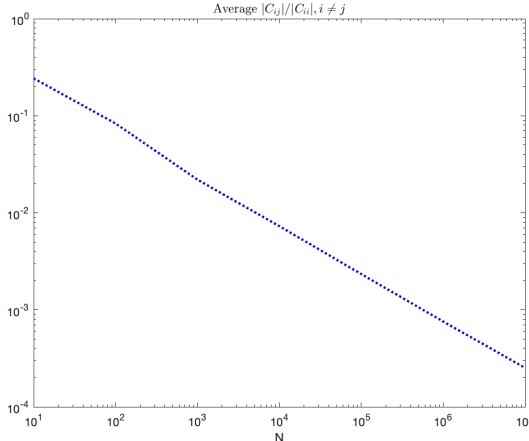

Figure 7: The ratio between the average magnitude of the diagonal elements $C_{ii}$ and off-diagonal elements $C_{ij}$ when $M = \alpha N$ with $\alpha = 0.5$.

Subsequently, under the Assumption 2, i.e., $\mathbf{A}\mathbf{A}^T$ is a diagonal matrix, each element $\tilde{n}_m \sim \mathcal{N}(\tilde{n}_m; 0, \sigma^2 + \beta_t^2 \left\|\mathbf{a}_m^T\right\|_2^2)$ of $\tilde{\mathbf{n}}$ will be independent to each other and thus from (38) we can obtain a closed-form solution for the likelihood distribution (we will refer it as a pseudo-likelihood due to the assumptions used) $p(\mathbf{y}|\hat{\mathbf{z}} = \mathbf{A}\tilde{\mathbf{x}})$ as follows

$$p(\mathbf{y}|\hat{\mathbf{z}} = \mathbf{A}\tilde{\mathbf{x}}) = \prod_{m=1}^{M} p\left(y_m \mid \hat{z}_m = \mathbf{a}_m^T \tilde{\mathbf{x}}\right) \tag{30}$$

where, from the definition of quantizer Q,

$$p\left(y_m \mid \hat{z}_m = \mathbf{a}_m^T \tilde{\mathbf{x}}\right) = p\left(l_{y_m} \leq \hat{z}_m + \tilde{n}_m < u_{y_m}\right) \tag{31}$$

$$= \Phi\left(\frac{-\hat{z}_m + u_{y_m}}{\sqrt{\sigma^2 + \beta_t^2 \left\|\mathbf{a}_m^T\right\|_2^2}}\right) - \Phi\left(\frac{-\hat{z}_m + l_{y_m}}{\sqrt{\sigma^2 + \beta_t^2 \left\|\mathbf{a}_m^T\right\|_2^2}}\right) \tag{32}$$

$$= \Phi\left(-\tilde{u}_{y_m}\right) - \Phi\left(-\tilde{l}_{y_m}\right) \tag{33}$$

where $\Phi(z) = \frac{1}{\sqrt{2\pi}} \int_{-\infty}^{z} e^{-\frac{t^2}{2}} dt$ is the cumulative distribution function of the standard normal distribution and

$$\tilde{u}_{y_m} = \frac{\mathbf{a}_m^T \tilde{\mathbf{x}} - u_{y_m}}{\sqrt{\sigma^2 + \beta_t^2 \left\|\mathbf{a}_m^T\right\|_2^2}}, \quad \tilde{l}_{y_m} = \frac{\mathbf{a}_m^T \tilde{\mathbf{x}} - l_{y_m}}{\sqrt{\sigma^2 + \beta_t^2 \left\|\mathbf{a}_m^T\right\|_2^2}}. \tag{34}$$

As a result, it can be calculated that the noise-perturbed pseudo-likelihood score $\nabla_{\tilde{\mathbf{x}}} \log p_{\beta_t}(\mathbf{y} \mid \tilde{\mathbf{x}})$ for the quantized measurements $\mathbf{y}$ in (2) can be computed as

$$\nabla_{\tilde{\mathbf{x}}} \log p_{\beta_t}(\mathbf{y} \mid \tilde{\mathbf{x}}) = \mathbf{A}^T \mathbf{G}(\beta_t, \mathbf{y}, \mathbf{A}, \tilde{\mathbf{x}}) \tag{35}$$

where $\mathbf{G}(\beta_t, \mathbf{y}, \mathbf{A}, \tilde{\mathbf{x}}) = [g_1, g_2, ..., g_M]^T \in \mathbb{R}^{M \times 1}$ with each element being

$$g_m = \frac{\exp\left(-\frac{\tilde{u}_{y_m}^2}{2}\right) - \exp\left(-\frac{\tilde{l}_{y_m}^2}{2}\right)}{\sqrt{\sigma^2 + \beta_t^2 \left\|\mathbf{a}_m^T\right\|_2^2} \int_{\tilde{l}_{y_m}}^{\tilde{u}_{y_m}} \exp\left(-\frac{t^2}{2}\right) dt}, \quad m = 1, 2, ..., M, \tag{36}$$

which completes the proof. $\qquad\square$

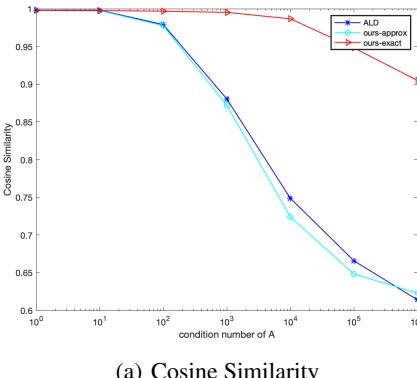
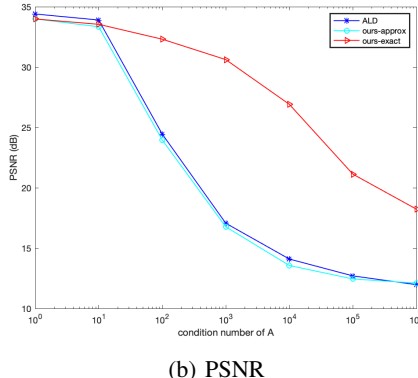

(a) Cosine Similarity

(b) PSNR

Figure 8: Averaged Cosine Similarity and PSNR of reconstructed MNIST images of ALD in Jalal et al. (2021a) and ours (with formula (18) and (16), respectively) when $M = 200, \sigma = 0.1$ for the different condition number of the matrix $\mathbf{A}$ is cond$(\mathbf{A}) = 1000$. It can be seen that our method with (16) significantly outperforms ALD in Jalal et al. (2021a) at high condition number while performing similarly at low condition number. Ours with diagonal approximation (18) is about the same as ALD as expected.

## D   PROOF OF COROLLARY 1.2

*Proof.* Similarly in the proof of Theorem 1, let us denote $\mathbf{z} = \mathbf{A}\mathbf{x}$. For each noise scale $\beta_t > 0$, under the Assumption 1, we can equivalently write

$$\mathbf{x} = \tilde{\mathbf{x}} + \beta_t \mathbf{w}, \tag{37}$$

where $\mathbf{w} \sim \mathcal{N}(\mathbf{0}, \mathbf{I})$. As a result, $\mathbf{z} = \mathbf{A}\mathbf{x} = \mathbf{A}(\tilde{\mathbf{x}} + \beta_t \mathbf{w}) = \mathbf{A}\tilde{\mathbf{x}} + \beta_t \mathbf{A}\mathbf{w}$. Then, in the case of the linear model, from (1), we obtain

$$\mathbf{y} = \mathbf{A}\tilde{\mathbf{x}} + \tilde{\mathbf{n}}, \tag{38}$$

where $\tilde{\mathbf{n}} = \mathbf{n} + \beta_t \mathbf{A}\mathbf{w}$. Since $\mathbf{n} \sim \mathcal{N}(\mathbf{0}, \sigma^2 \mathbf{I})$ and $\mathbf{w} \sim \mathcal{N}(\mathbf{0}, \mathbf{I})$ and are independent to each other, it can be concluded that $\tilde{\mathbf{n}}$ is also Gaussian with mean zero and covariance $\sigma^2 \mathbf{I} + \beta_t^2 \mathbf{A}\mathbf{A}^T$, i.e., $\tilde{\mathbf{n}} \sim \mathcal{N}(\tilde{\mathbf{n}}; \mathbf{0}, \sigma^2 \mathbf{I} + \beta_t^2 \mathbf{A}\mathbf{A}^T)$. Therefore, a closed-form solution for the likelihood distribution $p(\mathbf{y}|\hat{\mathbf{z}} = \mathbf{A}\tilde{\mathbf{x}})$ can be obtained as follows

$$p(\mathbf{y}|\hat{\mathbf{z}} = \mathbf{A}\tilde{\mathbf{x}}) = \frac{\exp\left(-\frac{1}{2}(\mathbf{y} - \mathbf{A}\tilde{\mathbf{x}})^T \left(\sigma^2 \mathbf{I} + \beta_t^2 \mathbf{A}\mathbf{A}^T\right)^{-1} (\mathbf{y} - \mathbf{A}\tilde{\mathbf{x}})\right)}{\sqrt{(2\pi)^M \det\left(\sigma^2 \mathbf{I} + \beta_t^2 \mathbf{A}\mathbf{A}^T\right)}}. \tag{39}$$

As a result, we can readily obtain a closed-form solution for the noise-perturbed pseudo-likelihood score $\nabla_{\tilde{\mathbf{x}}} \log p_{\beta_t}(\mathbf{y} \mid \tilde{\mathbf{x}})$ as follows

$$\nabla_{\tilde{\mathbf{x}}} \log p_{\beta_t}(\mathbf{y} \mid \tilde{\mathbf{x}}) = \mathbf{A}^T \left(\sigma^2 \mathbf{I} + \beta_t^2 \mathbf{A}\mathbf{A}^T\right)^{-1} (\mathbf{y} - \mathbf{A}\tilde{\mathbf{x}}). \tag{40}$$

Furthermore, if $\mathbf{A}\mathbf{A}^T$ is a diagonal matrix, the inverse of matrix $\left(\sigma^2 \mathbf{I} + \beta_t^2 \mathbf{A}\mathbf{A}^T\right)$ become trivial since it is a diagonal matrix with the $m$-th diagonal element being $\sigma^2 + \beta_t^2 \left\|\mathbf{a}_m^T\right\|_2^2$. After some simple algebra, we can obtain the equivalent representation in (18), which completes the proof. □

## E   COMPARISON WITH ALD IN JALAL ET AL. (2021A) IN THE LINEAR CASE

As shown in Corollary 1.2, in the special case without quantization, our results in Theorem 1 can be reduced to a form similar to the ALD in Jalal et al. (2021a). However, there are several different important differences. First, our results are derived in a principled way as noise-perturbed pseudo-likelihood score and admit closed-form solutions, while the results in Jalal et al. (2021a) are obtained

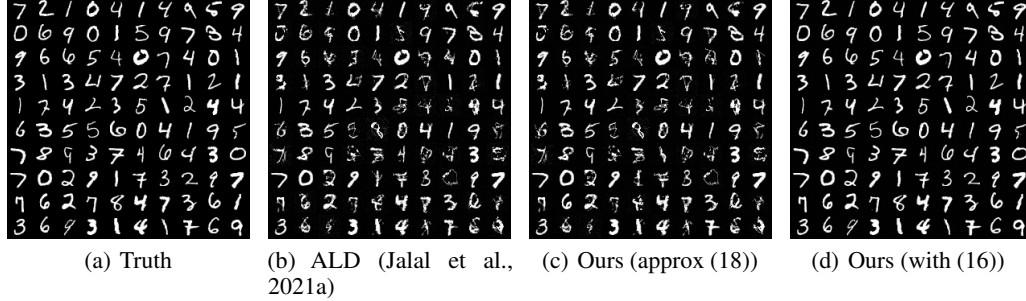

| (a) Truth | (b) ALD (Jalal et al., 2021a) | (c) Ours (approx (18)) | (d) Ours (with (16)) |

Figure 9: Tyical recovered MNIST images of ALD in Jalal et al. (2021a) and ours (with formula (18) and (16), respectively) when $M = 200, \sigma = 0.05$ and the condition number of matrix $\mathbf{A}$ is $\text{cond}(\mathbf{A}) = 1000$. It can be seen that our method with (16) significantly outperforms ALD in Jalal et al. (2021a), which performs about the same as ours with diagonal approximation (18).

heuristically by adding an additional hyper-parameter (and thus needs fine-tuning). Second, the results in Jalal et al. (2021a) are similar to an approximate version (18) of ours, which holds only when $\mathbf{A}\mathbf{A}^T$ is a diagonal matrix. For general matrices $\mathbf{A}$, we have a closed-form approximation (16). We compare our results with ALD (Jalal et al., 2021a) and the results are shown in Figure 8 and Figure 9. It can be seen that when at low condition number of $\mathbf{A}$ when $\mathbf{A}\mathbf{A}^T$ is approximately a diagonal matrix, our with diagonal approximation (18) performs similarly as (16), both of which are similar to ALD (Jalal et al., 2021a), as expected. However, when the condition number of $\mathbf{A}$ is large so that $\mathbf{A}\mathbf{A}^T$ is far from a diagonal matrix, our method with the (16) significantly outperforms ALD in Jalal et al. (2021a).

## F    DETAILED EXPERIMENTAL SETTINGS

In training NCSNv2 for MNIST, we used a similar training setup as Song & Ermon (2020) for Cifar10 as follows.

Training: batch-size: 128 n-epochs: 500000 n-iters: 300001 snapshot-freq: 50000 snapshot-sampling: true anneal-power: 2 log-all-sigmas: false.

Please refer to Song & Ermon (2020) and associated open-sourced code for details of training. For Cifar-10, CelebA, and FFHQ, we directly use the pre-trained models available in this Link.

When performing posterior sampling using the QCS-SGM in 1, for simplicity, we set a constant value $\epsilon = 0.0002$ for all quantized measurements (e.g., 1-bit, 2-bit, 3-bit) for MNIST, Cifar10 and CelebA. For the high-resolution FFHQ $256 \times 256$, we set $\epsilon = 0.00005$ for 1-bit and $\epsilon = 0.00002$ for 2-bit and 3-bit case, respectively. For all linear measurements for MNIST, Cifar10, and CelebA, we set $\epsilon = 0.00002$. It is believed that some improvement can be achieved with further fine-tuning of $\epsilon$ for different scenarios. For MNIST and Cifar-10, we set $\beta_1 = 50, \beta_T = 0.01, T = 232$; for CelebA, we set $\beta_1 = 90, \beta_T = 0.01, T = 500$; for FFHQ, we set $\beta_1 = 348, \beta_T = 0.01, T = 2311$ which are the same as Song & Ermon (2020). The number of steps $K$ in QCS-SGM for each noise scale is set to be $K = 5$ in all experiments. For more details, please refer to the submitted code.

## G    MULTIPLE SAMPLES AND UNCERTAINTY ESTIMATES

As one kind of posterior sampling method, QCS-SGM can yield multiple samples with different random initialization so that we can easily obtain confidence intervals or uncertainty estimates of the reconstructed results. For example, typical samples as well as mean and std are shown in Figure 10 for MNIST and CelebA in the 1-bit case.

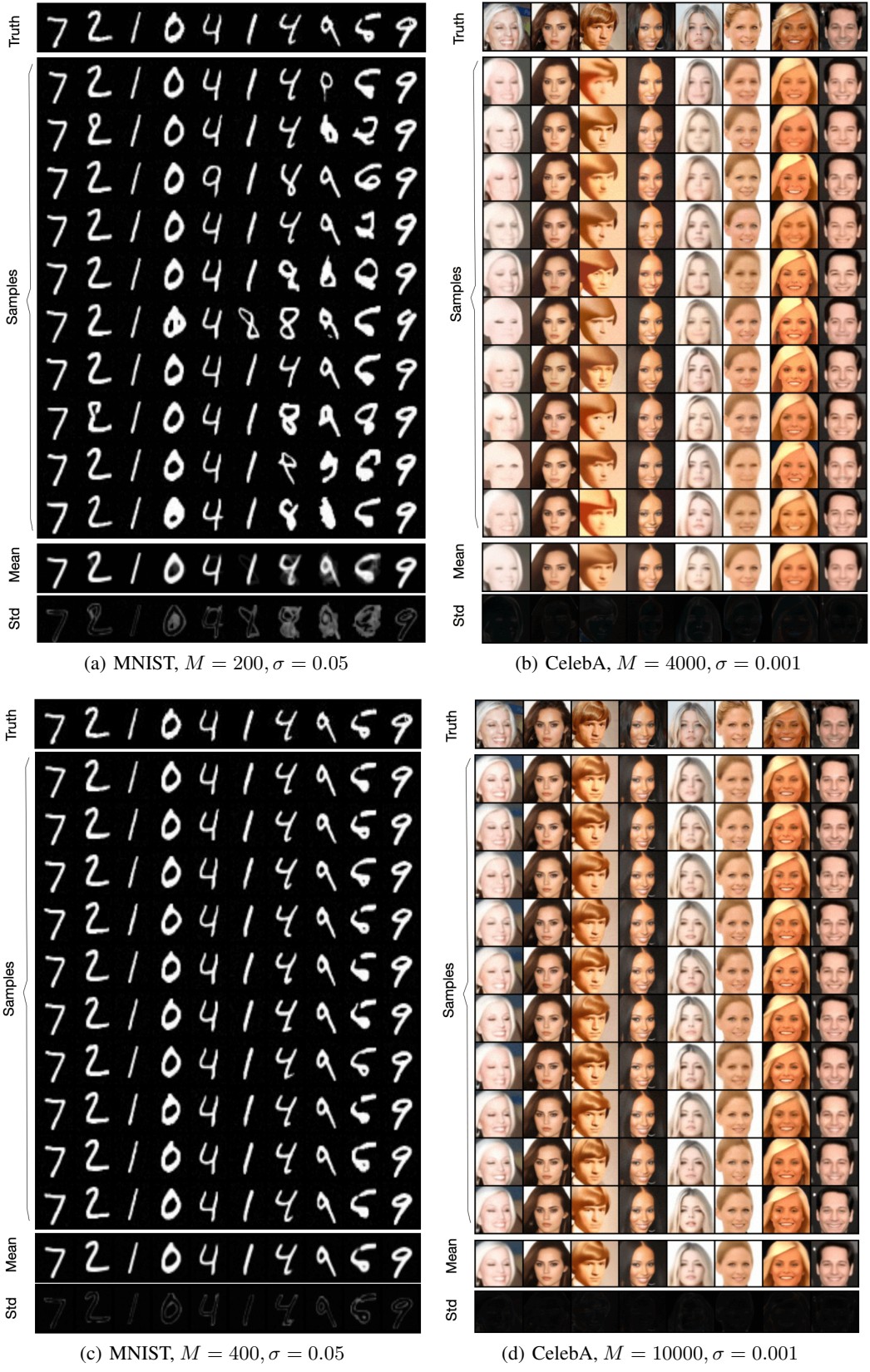

Figure 10: Multiple samples of QCS-SGM on MNIST ($M = 200, 400, \sigma = 0.05$) and CelebA datasets ($M = 4000, 1000, \sigma = 0.001$) from 1-bit measurements. The mean and std of the samples are also shown.

## H    ADDITIONAL RESULTS

Some additional results are shown in this section.

Figure 11 and Figure 12 show results with relatively large value of $M$ in the same setting as Figure 2 and Figure 4, respectively.

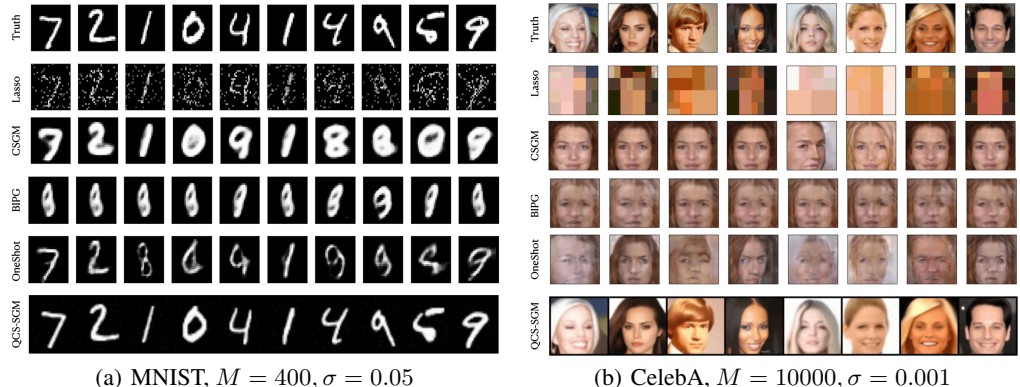

(a) MNIST, $M = 400, \sigma = 0.05$        (b) CelebA, $M = 10000, \sigma = 0.001$

Figure 11: Typical reconstructed images from 1-bit measurements on MNIST ($M = 400$) and CelebA ($M = 10000$).

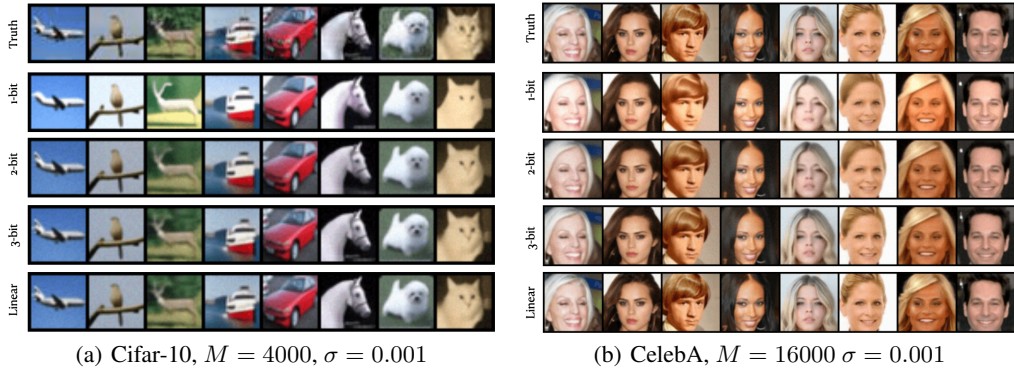

(a) Cifar-10, $M = 4000, \sigma = 0.001$        (b) CelebA, $M = 16000$ $\sigma = 0.001$

Figure 12: Results of QCS-SGM for Cifar-10 ($M = 4000$) and CelebA ($M = 16000$) images under different quantization bits.

Figure 13 shows the quantitative results of QCS-SGM for different quantization bits.

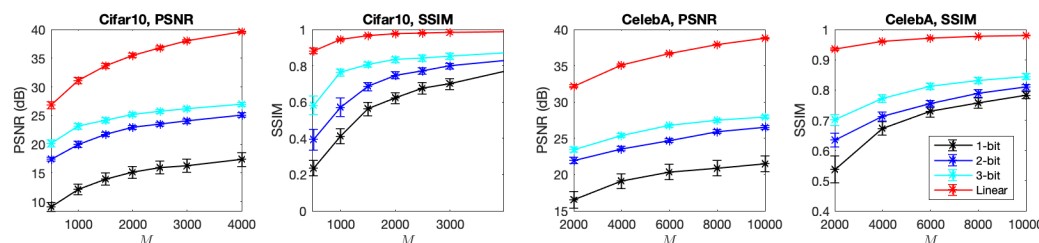

Figure 13: Quantitative results of QCS-SGM for different quantization bits, $\sigma = 0.001$.

Figure 14 and Figure 15 show the reconstructed images of QCS-SGM for Cifar10 and CelebA in the fixed budget case of $Q \times M = 3072$ and $Q \times M = 12288$, respectively.

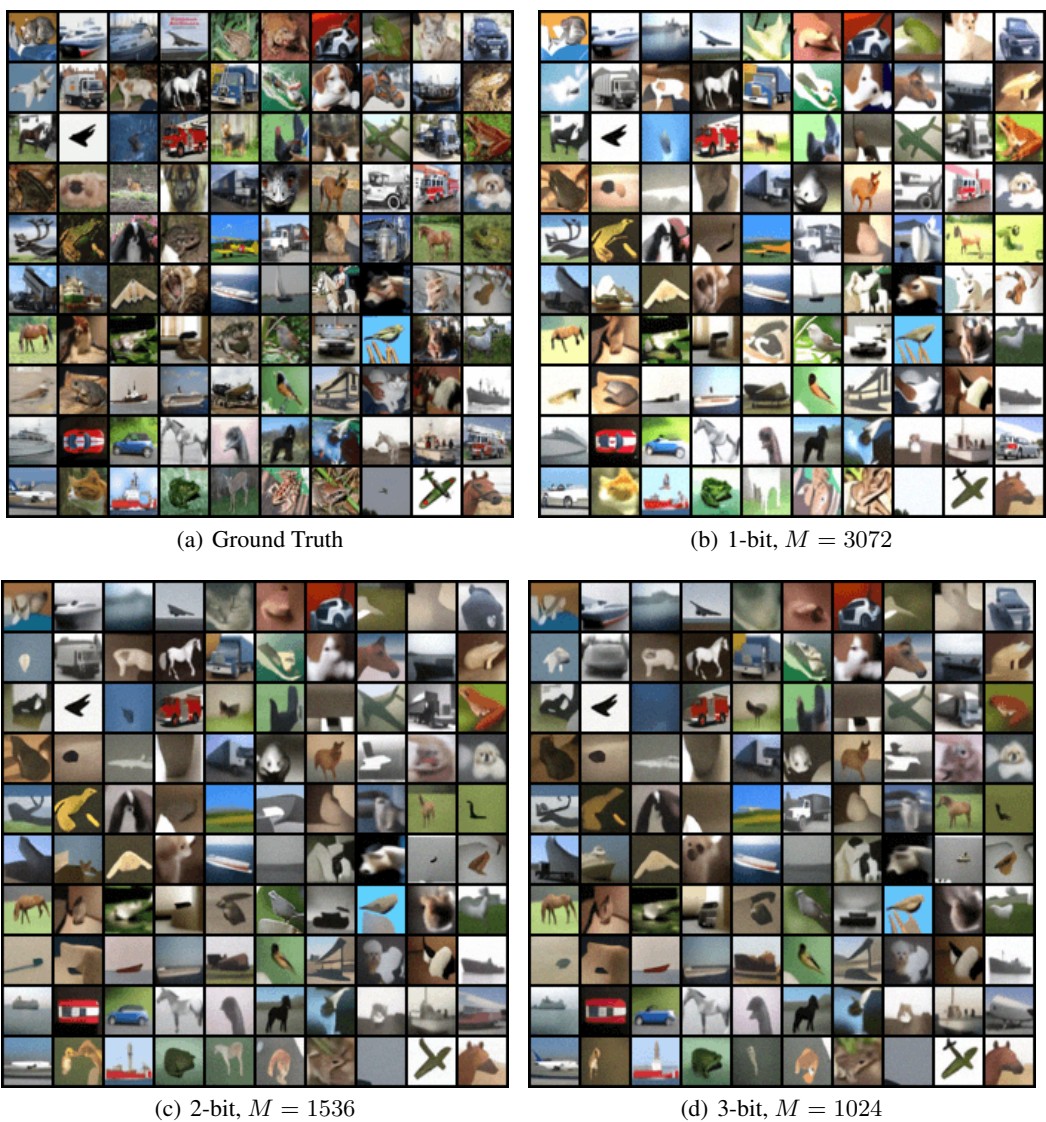

Figure 14: Reconstructed images of QCS-SGM for Cifar10 in the fixed budget case ($Q \times M = 3072$).

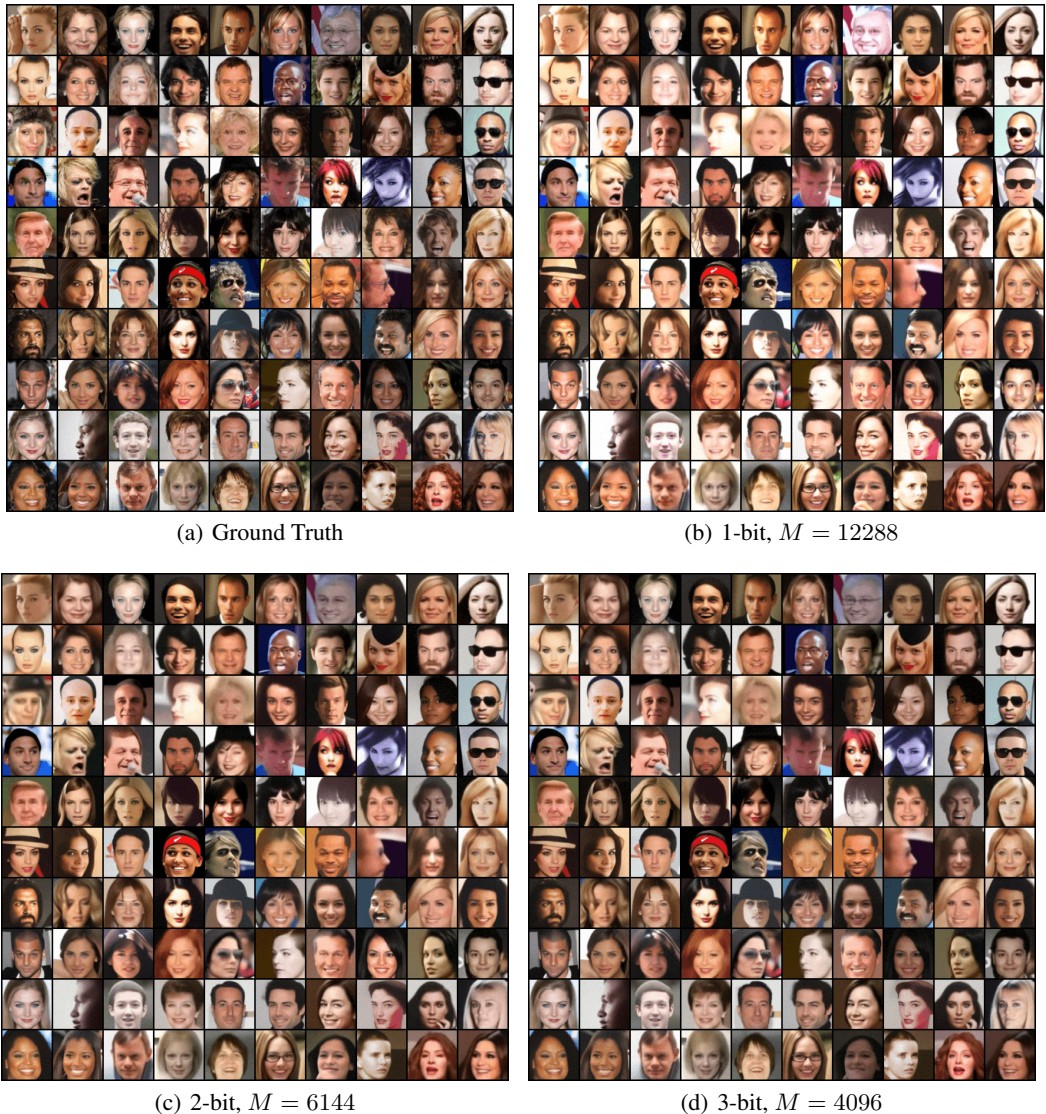

(a) Ground Truth

(b) 1-bit, $M = 12288$

(c) 2-bit, $M = 6144$

(d) 3-bit, $M = 4096$

Figure 15: Reconstructed images of QCS-SGM for CelebA in the fixed budget case ($Q \times M = 12288$) in the same setting as Table 1.

| Method | Cifar10 CS2 | | Cifar10 CS8 | | CelebA CS2 | | CelebA CS8 | |
|---|---|---|---|---|---|---|---|---|
| | SSIM ↑ | PSNR ↑ | SSIM ↑ | PSNR ↑ | SSIM ↑ | PSNR ↑ | SSIM ↑ | PSNR ↑ |
| QCS-SGM (ours) | **0.9712** | **34.33** | **0.8634** | **26.20** | **0.9694** | **36.80** | **0.9156** | **31.13** |
| Neumann networks (Gilton et al., 2019) | - | 33.83 | - | 25.15 | - | 35.12 | - | 28.38 |

Table 2: Quantitative comparison (PSNR (dB) and SSIM) of QCS-SGM with the Neumann networks in Gilton et al. (2019) for CS in the linear case, i.e., without quantization. Both Cifar10 $32 \times 32$ and CelebA $64 \times 64$ are considered. It can be seen that QCS-SGM outperforms the Neumann networks (Gilton et al., 2019) in all cases. As in Gilton et al. (2019). the values reported are the median across a test set of size 256. Reconstructed images of QCS-SGM are shown in Figure 16 - 19.

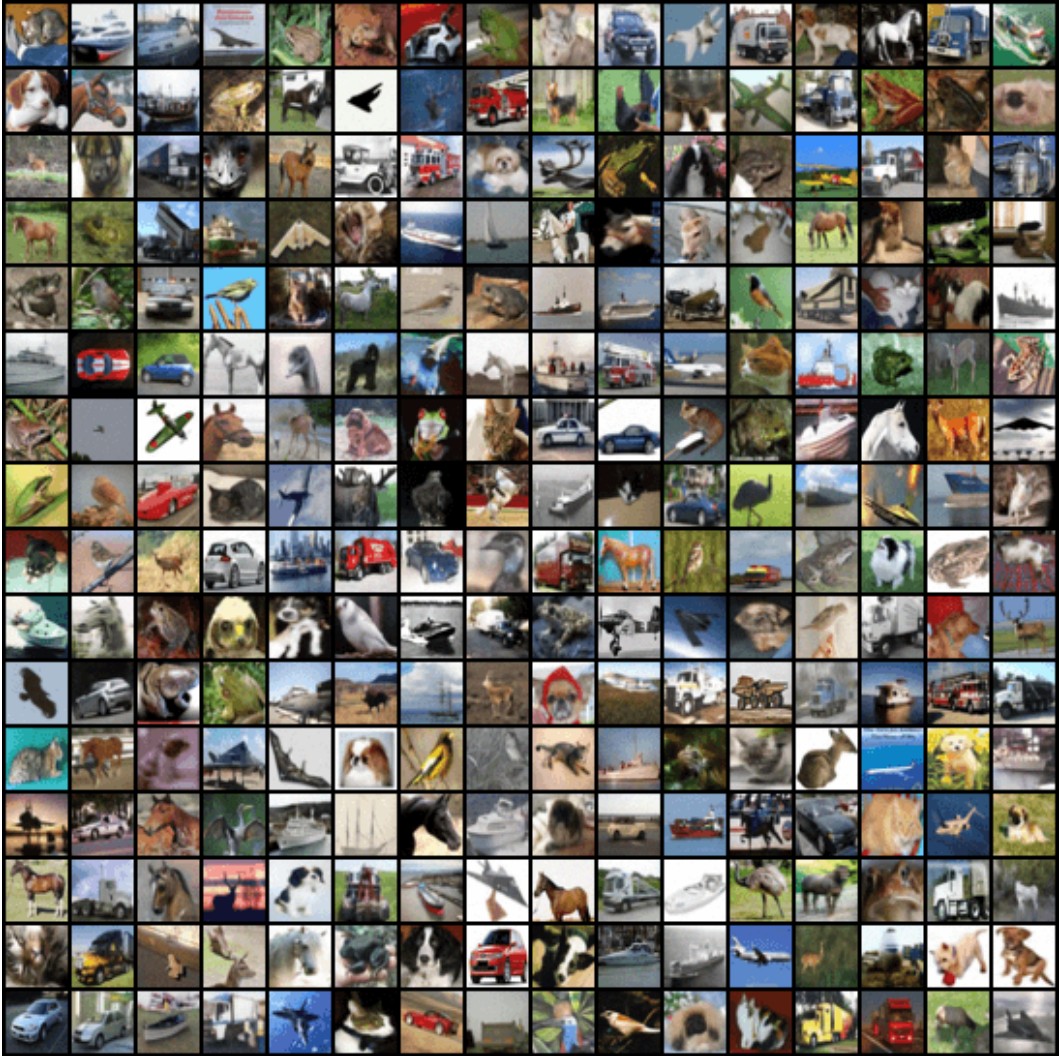

Figure 16: Results of QCS-SGM on Cifar10 for CS2, i.e., $M = 1536, N = 3072$. Noiseless case.

# I    COMPARISON WITH NEUMANN NETWORKS GILTON ET AL. (2019)

In the case of no quantization, we compare our method with one popular method called Neumann Networks (Gilton et al., 2019). The results for Cifar10 are shown in Table 2. It can be seen that our method outperforms Neumann Networks by a large margin. We do not show quantization comparisons for quantized CS since Neumann Networks does not support it.

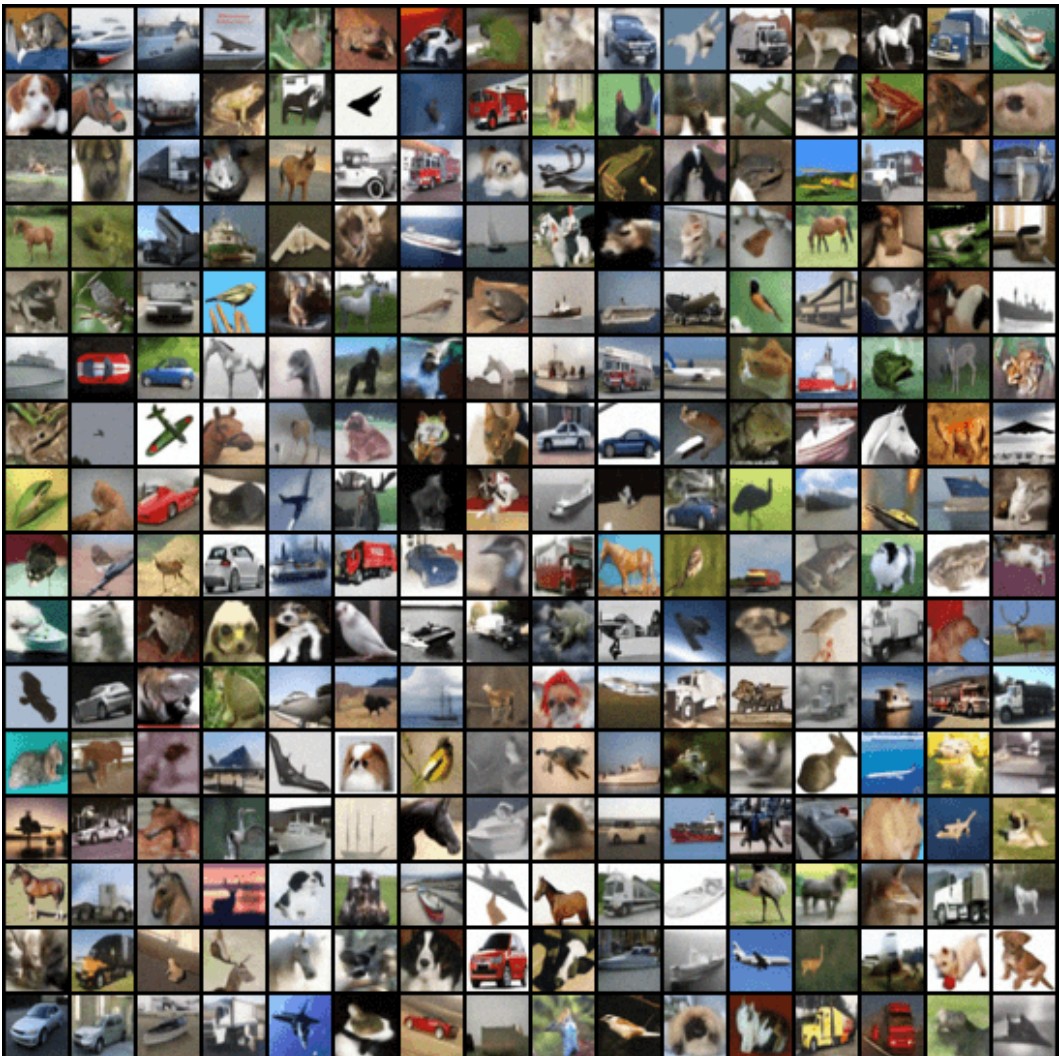

Figure 17: Results results of QCS-SGM on Cifar10 for CS8, i.e., $M = 1536, N = 3072$. Noiseless case.

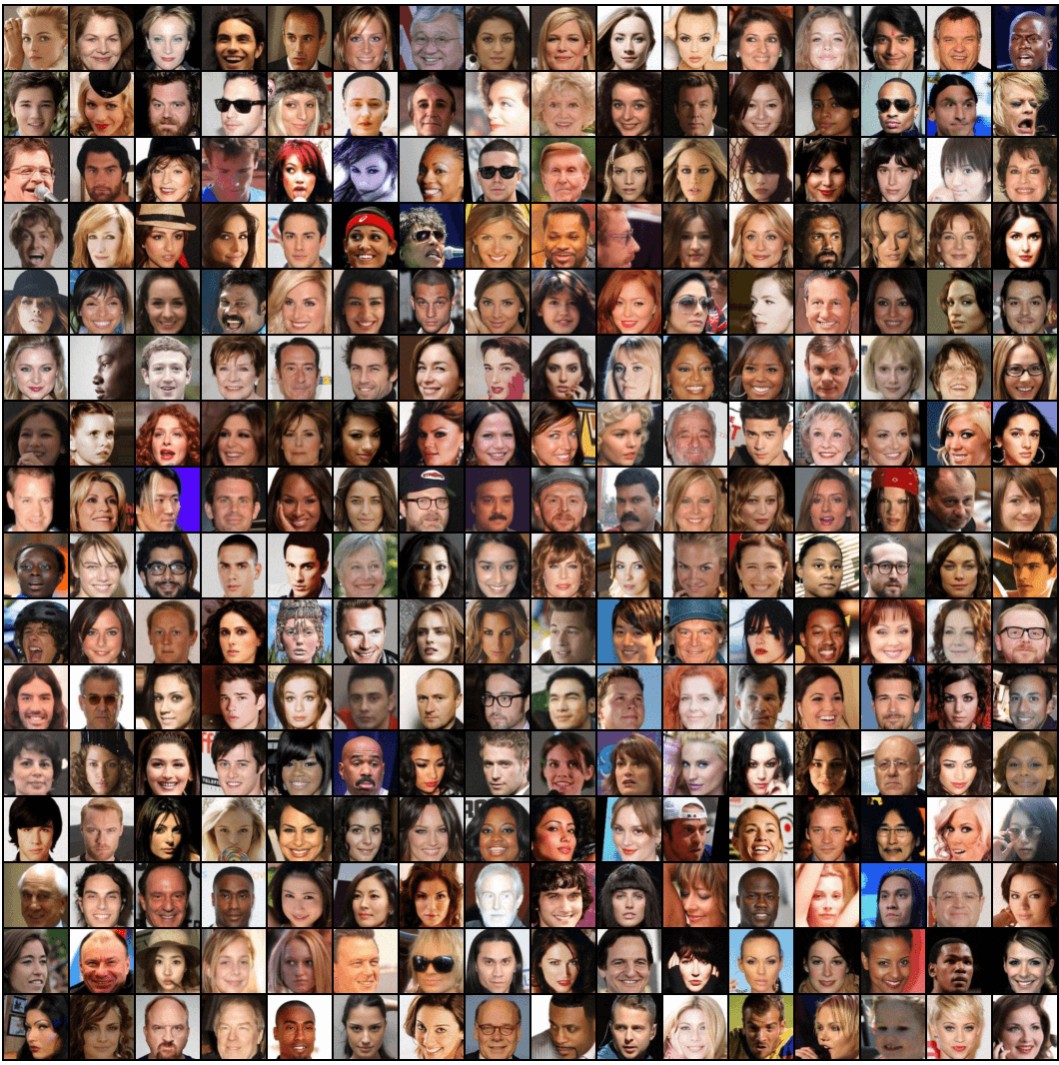

Figure 18: Results of QCS-SGM on CelebA for CS2, i.e., $M = 6144, N = 12288$. Noiseless case.

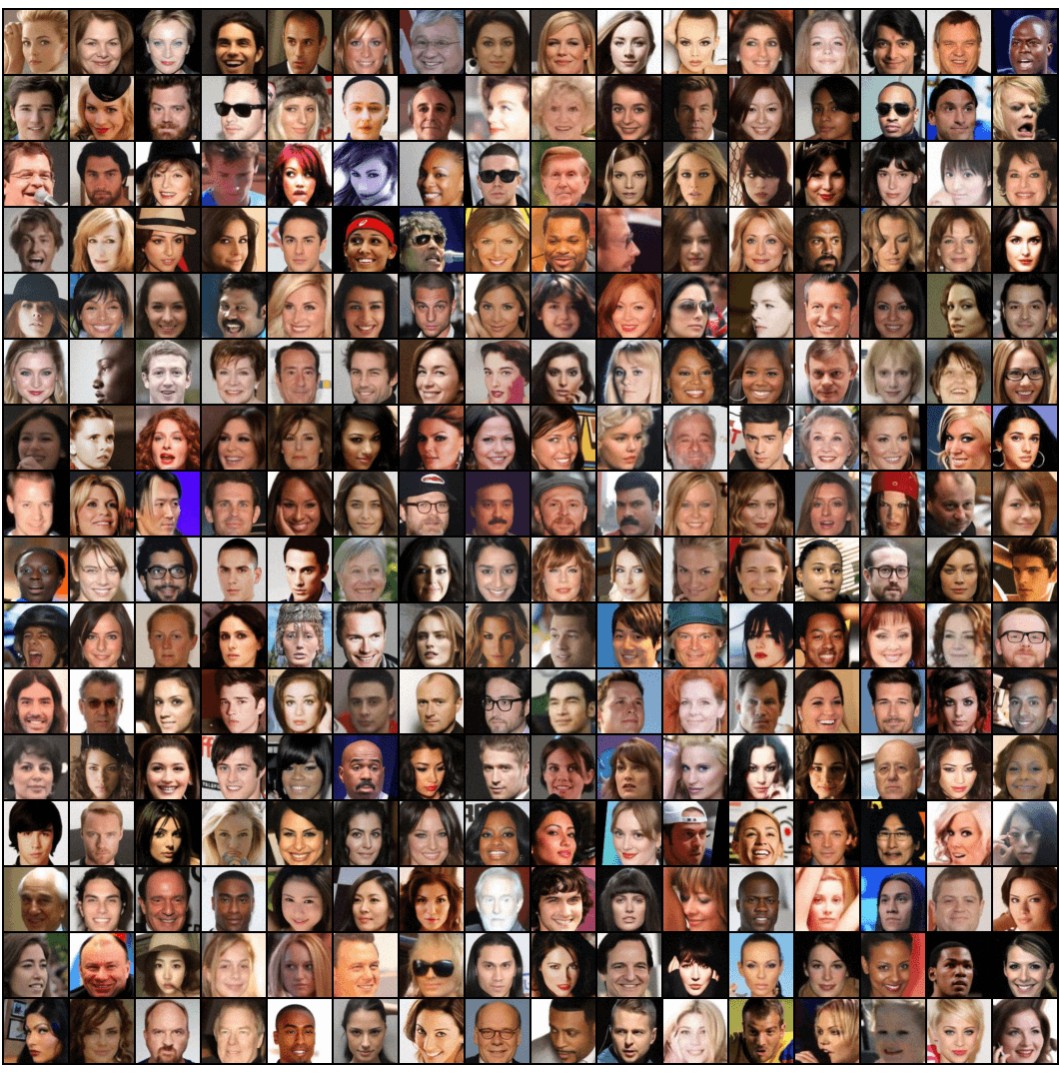

Figure 19: Results of QCS-SGM on CelebA for CS8, i.e., $M = 1536, N = 12288$. Noiseless case.

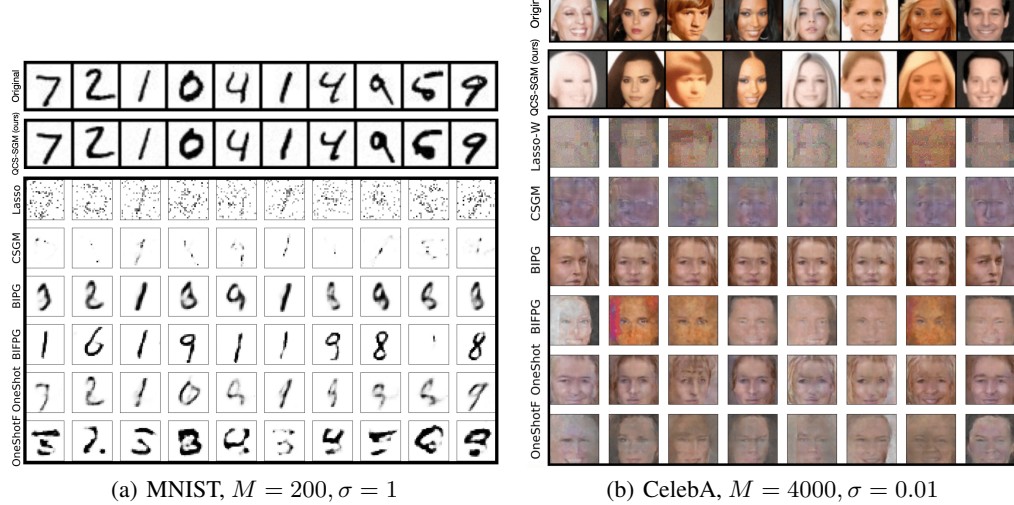

(a) MNIST, $M = 200, \sigma = 1$        (b) CelebA, $M = 4000, \sigma = 0.01$

Figure 20: Results of reconstructed images from 1-bit measurements on MNIST and CelebA images, following exactly the same setting as Liu & Liu (2022), i.e., $A_{ij} \sim \mathcal{N}(0,1)$, and $n_i \sim \mathcal{N}(0,\sigma^2)$.

## J    COMPARISON IN THE EXACTLY SAME SETTING AS LIU & LIU (2022)

Note that there is a slight difference in the modeling of measurement matrix $\mathbf{A}$ and noise $\mathbf{n}$ between ours and that in Liu & Liu (2022). In Liu & Liu (2022), it is assumed that $\mathbf{A}$ is an i.i.d. Gaussian matrix where each element $A_{ij}$ follows $A_{ij} \sim \mathcal{N}(0,1)$ and that $\mathbf{n}$ is an i.i.d. Gaussian with variance $\sigma^2$, i.e., $n_i \sim \mathcal{N}(0,\sigma^2)$. However, in practice, the measurement matrix $\mathbf{A}$ is usually normalized so that the norm of each column equals 1. As a result, in our setting, we assume that each element of i.i.d. Gaussian matrix follows $A_{ij} \sim \mathcal{N}(0,1/M)$. Mathematically, there is a one-to-one correspondence between the two settings, but the simulation setting is different due to the different measurement size $M$. As a result, for an exact comparison with results in Liu & Liu (2022), we also conducted experiments assuming exactly the same setting as Liu & Liu (2022), i.e., $A_{ij} \sim \mathcal{N}(0,1)$, and $n_i \sim \mathcal{N}(0,\sigma^2)$. The results are shown in Figures 20, 21, 22, 23.

An investigation of the effect of (pre-quantization) Gaussian noise is shown in Figure 23. Interestingly, the results in Figure 23 empirically demonstrate that QCS-SGM is robust to pre-quantization noise, i.e., similar performances have been achieved for a large range of noise, even when the noise is very large or approaching zero. Note that different from the comparing methods, the results of QCS-SGM in Figure 23 are not normalized to range $[0,1]$, which suggests that the "dithering" effect seems not apparent for QCS-SGM. A rigorous analysis of this is left as future work.

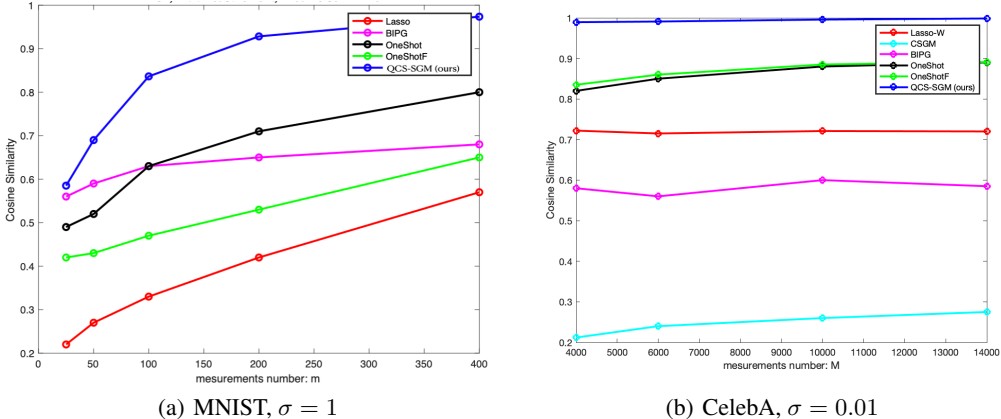

(a) MNIST, $\sigma = 1$      (b) CelebA, $\sigma = 0.01$

Figure 21: Quantitative comparisons based on cosine similarity for 1-bit MNIST and CelebA images, following exactly same setting as Liu & Liu (2022), i.e., $A_{ij} \sim \mathcal{N}(0,1)$, and $n_i \sim \mathcal{N}(0, \sigma^2)$.

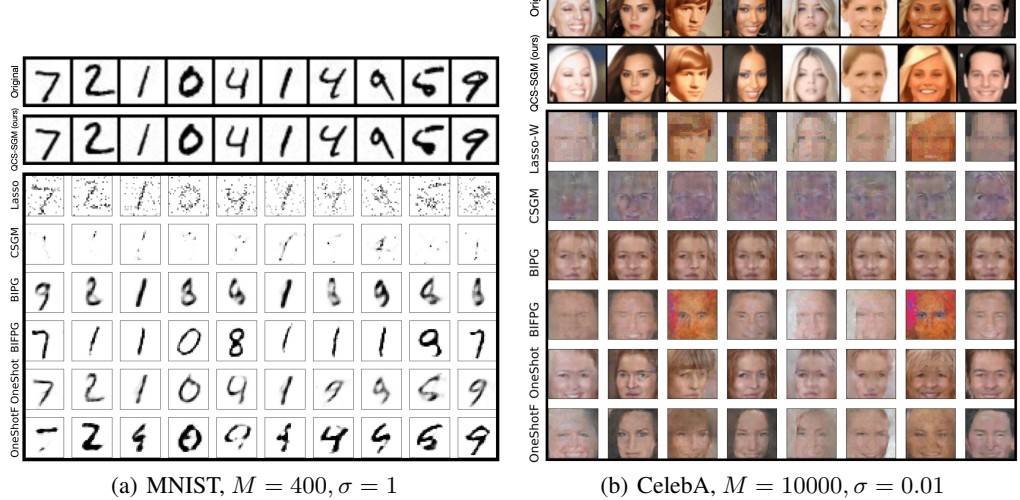

(a) MNIST, $M = 400, \sigma = 1$      (b) CelebA, $M = 10000, \sigma = 0.01$

Figure 22: Results of reconstructed images from 1-bit measurements on MNIST and CelebA images, following exactly the same setting as Liu & Liu (2022), i.e., $A_{ij} \sim \mathcal{N}(0,1)$, and $n_i \sim \mathcal{N}(0, \sigma^2)$.

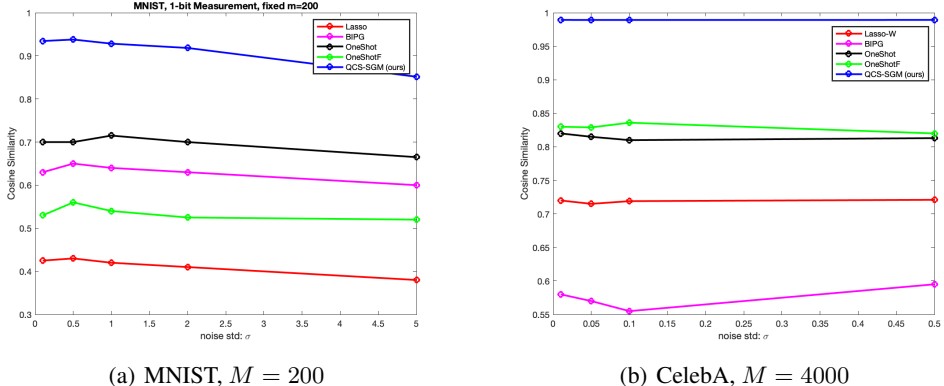

(a) MNIST, $M = 200$   (b) CelebA, $M = 4000$

Figure 23: Results of reconstructed images from 1-bit measurements on MNIST and CelebA images for different $\sigma$, following exactly the same setting as Liu & Liu (2022), i.e., $A_{ij} \sim \mathcal{N}(0, 1)$, and $n_i \sim \mathcal{N}(0, \sigma^2)$.

