# OpenReview forum: "Quantized Compressed Sensing with Score-Based Generative Models"
_ICLR.cc/2023/Conference — ICLR 2023 poster_

### Official Review · Reviewer_CEkX · 2022-10-21

**Confidence:** 3
**Correctness:** 3
**Technical Novelty And Significance:** 3
**Empirical Novelty And Significance:** 2
**Recommendation:** 6

**Clarity, Quality, Novelty And Reproducibility:**

The work is generally clear and the idea is novel. Reproducibility seems good from the textual description, and the authors pledged to release the code.

**Strength And Weaknesses:**

Strengths:
- Novel and interesting method to adapt score-based generative models to compressive sensing reconstruction in presence of quantization
- Good performance in the specific settings that are presented and on "toy" datasets like MNIST and CIFAR

Weaknesses:
- Experimental evaluation seems rather limited. Some of the alternative methods shown in the comparisons are not quite mainstream, while important and widely-used baselines for deep-learning CS image reconstruction are missing (for example, ISTA-net Zhang 2017, and Neumann networks Gilton 2019)
- Authors only report comparisons with other techniques for the 1-bit case but it is unclear how the method is positioned with respect to the state-of-the-art for a wider range of quantization rates
- For the purpose of quality comparisons of image reconstructions, I feel cosine similarity is not very useful. The most used metrics are just PSNR and SSIM. The space saved removing cosine similarity could be better used expanding the set of comparisons.
- Training SGM models is known to be very computationally expensive. The need for a pretrained one limits the applicability of the proposed method to a wider range of data for which no pretrained SGM is available, or it would be too expensive to train or not enough data are available. The authors need to discuss this limitation of the approach with respect to existing techniques, especially the ones not relying on generative models.
- Related to the previous point, results are presented on toy datasets for which pretrained SGMs are easily available, but it is unclear how the method generalizes to higher resolution images.

**Summary Of The Paper:**

The authors present a novel method for reconstruction from quantized compressive sensing measurements. The proposed method is a modification of score-based models to incorporate information about the forward measurement process and quantization noise. Overall, the method is interesting and generally well-explained. However, the experimental assessment seems somewhat incomplete. As a minor point, the quality of figures is poor and they are often hard to read.

**Summary Of The Review:**

The authors present a generally novel and interesting idea to use score-based models for CS reconstruction in presence of quantization. However, I feel the idea is not thoroughly validated from the experimental standpoint, making it unclear how it is positioned in the wider literature on CS reconstruction.

-----

Post-rebuttal: the authors significantly improved the original submission. I raised my score accordingly.

---

> ### Author Response · Authors · 2022-11-18
> **Author response to reviewer CEkX: Part I**
>
> We sincerely thank the reviewer for his/her constructive comments. We respond to the comments point by point as follows.
>
> $\bf{Q1}$:  Good performance in the specific settings that are presented and on "toy" datasets like MNIST and CIFAR.
>
> $\bf{A1}$: First, thank you for your positive comments. We would like to kindly point out that in the original manuscript, we not only provide results on "toy" datasets like MNIST and CIRAR10 but also more complex datasets like CelebA and FFHQ of size 64x64. In the rebuttal, we have also added results on high-resolution images of size 256x256 from FFHQ, as shown in Figure 1 of the revised manuscript. It can be seen that our proposed QCS-SGM can be generalized to high-resolution images quite well.
>
> $\bf{Q2}$: Experimental evaluation seems rather limited. Some of the alternative methods shown in the comparisons are not quite mainstream, while important and widely-used baselines for deep-learning CS image reconstruction are missing (for example, ISTA-net Zhang 2017, and Neumann networks Gilton 2019)
>
> $\bf{A2}$: Thank you for pointing out these alternative methods. We have included all of them in the references and added discussions in the main text. There are two main reasons why we did not compare with these deep-learning CS image reconstruction methods.
> - ISTA-net Zhang 2017  and Neumann networks Gilton 2019 are designed for unquantized measurements but not for the quantized CS, while in the current work we focus on quantized measurements (although ours applies to the unquantized case as well).
> - Both ISTA-net Zhang 2017 and Neumann networks Gilton 2019 are both supervised methods while our QCS-SGM is an unsupervised method. Specifically, the training of their methods requires pairs of $(x,y)$ while ours only need $x$. In particular, for ISTA-net, for different measurement models, even with different sizes M of measurements, they have to re-train the ISTA-net, which is very inflexible.  By contrast, for QCS-SGM, we do not need to re-train a model as long as we obtain a pre-trained diffusion model for $x$, i.e., it applies to any measurement model with any size M of measurements.
>
> In the revised manuscript, as suggested, we have provided comparisons with the Neumann networks Gilton 2019 on CIFAR10 and CelebA datasets for CS2 and CS8 in exactly the same setting as  Neumann networks Gilton 2019. The comparing results are shown in Appendix I (Table 2, Figures 16-19) and ours are much better than the Neumann networks Gilton 2019 even in the linear case. In the quantized case, as Neumann networks ignore the quantization, it is expected the gap between  Neumann networks and QCS-SGM will become larger in the quantized case. Nevertheless, as they did not provide the pre-trained models in their open-sourced code, it is much difficult for us to implement it in such a short time, so we did not report the results of Neumann networks for the quantized case.
>
> $\bf{Q3}$: Authors only report comparisons with other techniques for the 1-bit case but it is unclear how the method is positioned with respect to the state-of-the-art for a wider range of quantization rates.
>
> $\bf{A3}$: Thank you for the comment. In the revised manuscript, we have added results of different methods for multi-bit cases under a  "fixed budget basis" setting as suggested by Reviewer CmCq. As can be seen from Table 1, our method still achieves significantly superior performances in the multi-bits case.  Please refer to Table 1 in the revised manuscript (More results are shown in Figure 14, 15 in Appendix H). Moreover, in the case of infinite quantization rates, i.e., unquantized linear case, we have also added comparisons with the suggested Neumann networks Gilton 2019) in Appendix I (Table 2, Figures 16-19), and again, our proposed QCS-SGM outperforms the Neumann networks Gilton 2019 for both CIfar10 and CelebA datasets in various settings. As a result, the proposed method is advantageous in a much wider range of quantization rates ranging from the 1-bit to infinite-bit (linear case).
>
> $\bf{Q4}$: For the purpose of quality comparisons of image reconstructions, I feel cosine similarity is not very useful. The most used metrics are just PSNR and SSIM. The space saved by removing cosine similarity could be better used to expand the set of comparisons.
>
> $\bf{A4}$: Thank you very much for this suggestion. We have deleted the results of cosine similarity in the main text and only keep them in the Appendix J.

---

> ### Author Response · Authors · 2022-11-18
> **Author response to reviewer CEkX: Part II**
>
> $\bf{Q5}$: Training SGM models are known to be very computationally expensive. The need for a pre-trained one limits the applicability of the proposed method to a wider range of data for which no pretrained SGM is available, or it would be too expensive to train or not enough data are available. The authors need to discuss this limitation of the approach with respect to existing techniques, especially the ones not relying on generative models.
>
> $\bf{A5}$: We agree that our method relies on the availability of a pre-trained diffusion model or the ability to train such a diffusion model. Nevertheless, this is a common disadvantage of all methods based on generative models. While it is true that diffusion models are expensive to train currently, it is believed that in the future more efficient training methods can be proposed.
>
> As suggested, we have added more detailed discussions on the limitation of our method in the Conclusion (section 5) of the  revised manuscript in the rebuttal.
>
> $\bf{Q6}$: Related to the previous point, results are presented on toy datasets for which pre-trained SGMs are easily available, but it is unclear how the method generalizes to higher-resolution images.
>
> $\bf{A6}$:  Thank you for the suggestion. We would like to kindly point out that, apart from low-resolution datasets MNIST and Cifar10, we have provided results on relatively high-resolution images of size 64x64 from both FFHQ and CelebA datasets in the original manuscript. To further demonstrate its ability to generalize to higher resolution images, in the revised manuscript, we have added results on 256x256 FFHQ, as shown in the rebuttal.  As can be seen in Figure 1, the proposed QCS-SGM achieves surprisingly good results for 256x256 high-resolution images, which demonstrates the excellent generalization ability of our method to high-resolution images.
>
>
> $\bf{Q7}$: The authors present a generally novel and interesting idea to use score-based models for CS reconstruction in presence of quantization. However, I feel the idea is not thoroughly validated from the experimental standpoint, making it unclear how it is positioned in the wider literature on CS reconstruction.
>
> $\bf{A7}$: Thank you for the comments. In the revised manuscript, we have added substantial experiments to further validate the proposed method as suggested, for example, adding experiments on high-resolution 256x256 images, and comparisons with Neumann networks Gilton 2019, as well as a comparison of different algorithms under different multi-bits in a fixed budget setting, etc. Please refer to the revised manuscript for detail. We hope that the rebuttal could address your concern and sincerely hope that you could re-evaluate our work based on these revisions.

---

> > ### Comment · Reviewer_CEkX · 2022-11-28
> > **Thank you for your work**
> >
> > After reading the rebuttal and the revised paper, I think the quality of the submission has improved and many outstanding points have been addressed. I raised my score accordingly.

---

> > > ### Author Response · Authors · 2022-11-28
> > > **Thank you for your positive feedback.**
> > >
> > > We are happy that your concerns have been addressed. Thank you for raising the score.

---

### Official Review · Reviewer_CmCq · 2022-10-23

**Confidence:** 4
**Correctness:** 3
**Technical Novelty And Significance:** 3
**Empirical Novelty And Significance:** 4
**Recommendation:** 8

**Clarity, Quality, Novelty And Reproducibility:**

There are a handful of points that can be clarified/corrected:

Theorem 1 is somewhat confusing as phrased. Can one instead start with the assumption of orthogonality for the rows of A, and then make the "if gaussian then closed form solution" statement after?

In Corollary 1.2, it is not clear what a_m corresponds to in (17) - it appears at first look that this equation is meant to be for entry m of the score?

Page 5, typo "expliciitly"
Algorithm 1, suggest changing the reference to eq (25) to be (11-13) so that the reader does not have to delve into the appendix.

**Strength And Weaknesses:**

The numerical results use real-world datasets of varying complexity and show significant performance improvements vs. other generative model-based approaches to quantized CS recovery. However, I would argue that the use of Daubechies-1 (i.e., Haar) wavelets for images does not provide a fair comparison, as other wavelet or patch DCT bases are widely known to be better suited for images. Other results show good scaling to multi-bit quantization and out-of-distribution samples.

The method opens some questions that the authors may want to briefly address, or consider in the future:

It would be interesting to comment on the robustness of these results when the rows of the measurement matrix are only approximately orthogonal, or perhaps validate it numerically.

It would be interesting to consider the effect of noise in the recovery, and how the different methods would perform on a "fixed budget basis" setting - e.g, M one-bit measurements vs. M/B B-bit measurements.

**Summary Of The Paper:**

The paper proposes an approach for signal recovery in quantized compressed sensing (CS) that leverages score-based generative models. This addresses a specific setting for the first time, as previous approaches to SGM-based CS do not assume quantization, and quantized CS approaches with generative models have used other options such as GANs and variational autoencoders. A new score is proposed that leverages knowledge of the measurements when assessing the likelihood of a particular candidate signal.

Sample candidates for the recovered signal are generated using a Angevin dynamics formulation where the probability term becomes conditioned on the measurements observed. A Bayes rule is leveraged to allow for the score training process to be independent of the measurement matrix.

The paper sets up a recovery method that leverages a score-based generative model and derives a closed form solution under an orthogonal-row measurement matrix assumption. This result is specialized to the one-bit and non-quantized cases.


**Summary Of The Review:**

An interesting approach to quantized CS recovery with good analytical treatment and compelling experimental results.

---

> ### Author Response · Authors · 2022-11-18
> **Author response to reviewer CmCq**
>
> We sincerely thank the reviewer for his/her constructive and positive comments. We respond to the comments point by point as follows.
>
> $\bf{Q1}$: However, I would argue that the use of Daubechies-1 (i.e., Haar) wavelets for images does not provide a fair comparison, as other wavelet or patch DCT bases are widely known to be better suited for images.
>
> $\bf{A1}$: Thank you for pointing out this. Previously, we follow the same setting as Liu & Liu 2022 to use Lasso-W.  in the revised manuscript, we have replaced Lasso-W with Lasso-DCT as suggested.
>
> $\bf{Q2}$: It would be interesting to comment on the robustness of these results when the rows of the measurement matrix are only approximately orthogonal, or perhaps validate it numerically.
>
> $\bf{A2}$: Thank you for the comments. We verify the assumption in Appendix B when the measurement matrix is i.i.d. Gaussian so that row-orthogonality only holds approximately. Simply speaking, for i.i.d. Gaussian matrix $A$, the off-diagonal elements of the matrix $AA^T$ will concentrate to zero in the large limit $N\to \infty$, for which we also validated using numerical simulations. Note that for moderate $N$, e.g., for the small-size MNIST dataset, matrix $AA^T$ might be a bit far from diagonal matrices, however, our simulation results demonstrate that such an approximation still yields surprisingly good results. It is conjectured that this is because, although the specific values of the gradient of the log-likelihood, i.e., likelihood scores are not exact, the approximated score provides the correct information of the direction.
>
> $\bf{Q3}$: It would be interesting to consider the effect of noise in the recovery, and how the different methods would perform on a "fixed budget basis" setting - e.g, $M$ one-bit measurements vs. $M/B$ $B$-bit measurements.
>
> $\bf{A3}$: Thank you for the comments and suggestions. In the original submission, we considered the effect of noise in the Appendix, as shown in Figure 23 of Appendix J in the revised manuscript. In the rebuttal, we have added results in the  "fixed budget basis" setting as suggested, please refer to Table 1 in the revised manuscript. In addition, more results of the reconstructed images in the  "fixed budget basis" setting are shown in Figure 14 and Figure 15 of Appendix H.
>
> $\bf{Q4}$: Theorem 1 is somewhat confusing as phrased. Can one instead start with the assumption of orthogonality for the rows of $A$, and then make the "if gaussian then closed form solution" statement after?
>
> $\bf{A4}$: Thank you very much for the comments and suggestions. We have rephrased Theorem 1 in a clearer and more rigorous form by introducing two assumptions, one is the assumption of orthogonality for the rows of $A$ as suggested, and the other is the uninformative assumption of the prior. Verifications of the two assumptions are also provided in Appendix A and B, respectively. Please refer to the revised manuscript for details.
>
> $\bf{Q5}$: In Corollary 1.2, it is not clear what ${\bf{a}}_m$ corresponds to in (17) - it appears at first look that this equation is meant to be for entry $m$ of the score?
>
> $\bf{A5}$: Thank you for pointing out this typo. a_m denotes the m-th row of matrix $A$ and we have added definitions of ${\bf{a}}_m$ in the revised version (in between (12) and (13) where it is first used). Yes, you are right, (17) was initially intended to describe the $m$-th entry of the score but there was a typo.  We have corrected this typo and described (17) in a clearer form.
>
>
> $\bf{Q6}$: Page 5, typo "expliicitly" Algorithm 1, suggest changing the reference to eq (25) to be (11-13) so that the reader does not have to delve into the appendix.
>
> $\bf{A6}$: Thank you for pointing out this. We have corrected them as suggested.

---

### Official Review · Reviewer_DMn5 · 2022-10-24

**Confidence:** 4
**Correctness:** 3
**Technical Novelty And Significance:** 3
**Empirical Novelty And Significance:** 3
**Recommendation:** 6

**Clarity, Quality, Novelty And Reproducibility:**

This paper is well-written, and the main idea of this work is natural and clearly presented. This work may seem to be incremental compared to Jalal et al., 2021a and Liu & Liu, 2022.

As for reproducibility, I hope that the code has been submitted along with the main paper, instead of saying that "Our code will be open-sourced after acceptance".

**Details Of Ethics Concerns:**

NA.

**Strength And Weaknesses:**

Strength:

As the authors mentioned, their work is the first one that utilizes SGM quantized CS. Such an idea is interesting, and the experimental results are promising.

Weaknesses:

(a). SGM has been used for linear CS in some recent works, including Jalal et al., 2021a;b and

Daras, G., Dagan, Y., Dimakis, A. and Daskalakis, C., 2022, June. Score-Guided Intermediate Level Optimization: Fast Langevin Mixing for Inverse Problems. In International Conference on Machine Learning (pp. 4722-4753). PMLR.

I hope the authors can provide more discussions on these closely relevant works and provide intuition on why the QCS-SGM can outperform the methods proposed in these works. Ideally, similar to Jalal et al., 2021a, also perform the experiments for some MRI datasets.

(b). In Theorem 1, the assumption that $\mathbf{A}\mathbf{A}^T$ is diagonal is restrictive. Although this assumption holds approximately for some popular measurement matrices such as i.i.d. Gaussian and discrete cosine transform matrices, it is more desired to provide guarantees for these measurement matrices directly (i.e., for the case that $\mathbf{A}\mathbf{A}^T$ is approximately diagonal ).

**Summary Of The Paper:**

Motivated by the power of score-based generative models (SGM), this work studies quantized compressed sensing with score-based generative models and proposed an unsupervised data-driven approach that is denoted by QCS-SGM. One key to QCS-SGM is proposing an annealed likelihood score (noise-perturbed likelihood score), which incorporates information from the measurements while performing the posterior sampling. In particular, if $\mathbf{A}\mathbf{A}^T$ is a diagonal matrix, QCS-SGM reduces to a form similar to that of Jalal et al. (2021a), and corresponding guarantees are provided in Theorem 1 and Corollary 1.1. Numerical results are provided to show the superiority of QCS-SGM.

**Summary Of The Review:**

Although this work seems to be incremental compared to some closely relevant prior works, the main idea is interesting and the experimental results are promising. Therefore, I am inclined to acceptance.

---

> ### Author Response · Authors · 2022-11-18
> **Author response to reviewer DMn5: Part I**
>
> We sincerely thank the reviewer for his/her constructive comments. We respond to the comments point by point as follows.
>
> $\bf{Q1}$: SGM has been used for linear CS in some recent works, including Jalal et al., 2021a;b and  Daras, G., Dagan, Y., Dimakis, A. and Daskalakis, C., 2022, June. Score-Guided Intermediate Level Optimization: Fast Langevin Mixing for Inverse Problems. In International Conference on Machine Learning (pp. 4722-4753). PMLR.
> I hope the authors can provide more discussions on these closely relevant works and provide intuition on why the QCS-SGM can outperform the methods proposed in these works.
>
> $\bf{A1}$: Thank you for pointing out the interesting related work Daras et al. 2022. We have added it in the related work part of our revised manuscript with additional discussions on the intuition that why ours outperforms previous ones. Note that one big difference of Daras et al. 2022 from alal et al., 2021a;b and ours is that they do posterior sampling in the "latent space" of a pre-trained generative model, e.g., StyleGAN-2. Simply speaking, there are two main reasons why QCS-SGM outperforms previous methods.
>
>   - QCS-SGM explicitly considers quantization while previous methods, including Jalal et al., 2021a;b, Daras et al. 2022, only focus on linear measurements without quantization. This is the most important reason why QCS-SGM outperforms previous ones. It is worth pointing out that, as shown in our manuscript,  the extension of the linear case to non-linear quantized measurements is far from trivial since it is intractable in general.
>    - In calculating the likelihood score of $p(y|x_t)$, previous methods such as Jalal et al., 2021a;b directly compute the score of $p(y|x_0)$ by assuming that $x_t = x_0$, which is apparently not the case for $t>0$ and thus leads to an inaccurate approximation of the likelihood score, thus resulting in performance degradation. In fact, in the original Jalal et al., 2021b, they have to add an additional annealing term to compensate for their inaccurate approximation (otherwise, it does not work well), which is in fact a special case of our generalized results, as illustrated in Remark 3 of our manuscript. This is the second reason why QCS-SGM even outperforms Jalal et al., 2021a;b in the linear case for general matrices A. We have provided a comparison with Jalal et al., 2021a in the linear case on MNIST, as shown in Figure 8 and Figure 9 of Appendix E to illustrate this difference.
>
> $\bf{Q2}$: Ideally, similar to Jalal et al., 2021a, also perform the experiments for some MRI datasets.
>
> $\bf{A2}$: Thank you for this suggestion.  Unfortunately, due to a lack of time, we do not provide results. Nevertheless, we would like to kindly point out that there is in fact no need to conduct experiments on MRI to compare with Jalal et al., 2021a. The reasons are as follows:
>    - In Corollary 1.2 of our manuscript, we have provided a general result for the likelihood score in the linear case, and demonstrated in Remark 3 that the result of Jalal et al., 2021a is a special case of ours under the row-orthogonality assumption of A. Interestingly, for MRI, the measurement matrix A is exactly row-orthogonal since it is a Fourier transform matrix, as shown in Jalal et al., 2021a. Therefore, from Corollary 1.2 and Remark 3,  in the linear case, the algorithm Jalal et al., 2021a has the same form as ours and should be expected to achieve the same performance regardless of the datasets.
>    - In Appendix E, we provided a comparison with Jalal et al., 2021a in the linear case on MNIST, as shown in Figure 8 and Figure 9. Although the dataset is simple and small compared to MRI, it is sufficient to demonstrate and support our theoretical analysis in Corollary 1.2 and Remark 3.
>    -  While in principle it is easy to apply our algorithm to MRI, it is kind of difficult in practical coding since in MRI all the signals, sensing matrix, noise, and measurements become complex-valued. As a result, we have to modify our code to adapt to the complex-valued data, which is not that easy in such a short time. Indeed, such a successful application in Jalal et al., 2021a has resulted in a NeurPS publication. In fact, we have had a try on MRI using their open-sourced code, however, we have some difficulty in the second step as described in the README file: Install BART for sensitivity map estimation due to the sudo operation. We guess that this might be due to our improper environment setting.

---

> ### Author Response · Authors · 2022-11-18
> **Author response to reviewer DMn5: Part II**
>
> $\bf{Q3}$: In Theorem 1, the assumption that $AA^T$ is diagonal is restrictive. Although this assumption holds approximately for some popular measurement matrices such as i.i.d. Gaussian and discrete cosine transform matrices, it is more desired to provide guarantees for these measurement matrices directly (i.e., for the case that $AA^T$  is approximately diagonal ).
>
> $\bf{A3}$: Thank you for the comments. We have added a verification in Appendix B of the revised manuscript. Simply speaking, for i.i.d. Gaussian matrix $A$, the off-diagonal elements of the matrix $AA^T$ will concentrate to zero in the large limit, which we also validated using numerical simulations.
> Besides, we fully agree that the assumption that $AA^T$ is diagonal is indeed restrictive. Nevertheless, here we would like to kindly remark a bit that in the field of CS, this does not impose a very severe restriction since the sensing matrices A in CS are carefully designed and the most popular (or useful) matrices satisfy this assumption exactly, for example, discrete Fourier transform (DFT) in MRI, discrete cosine transform (DCT), Hadamard matrices, as well as other kinds of row-orthogonal matrices [R1]. Notably, it has been theoretically demonstrated in [R1] that, in various practical application scenarios, the row-orthogonal matrices are mostly desired with superior performance and easy implementation [R1].
> As suggested, we have explicitly acknowledged the limitation of our algorithm for such an assumption and leave the design for general matrices as an important future work.
>
> [R1] M. Vehkaperä, Y. Kabashima and S. Chatterjee, "Analysis of Regularized LS Reconstruction and Random Matrix Ensembles in Compressed Sensing," in IEEE Transactions on Information Theory, vol. 62, no. 4, pp. 2100-2124, April 2016, doi: 10.1109/TIT.2016.2525824.
>
> $\bf{Q4}$: This work may seem to be incremental compared to Jalal et al., 2021a and Liu & Liu, 2022.
>
> $\bf{A4}$: While ours are closely related to Jalal et al., 2021a and Liu & Liu, 2022, we believe that this work is more than incremental compared to them, as explained below:
>
>  - Compared with Jalal et al., 2021a, first of all, Jalal et al., 2021a is restricted to the linear case while our method generalizes Jalal et al., 2021a from the linear case to nonlinear quantization, which is both important and highly non-trivial. Moreover, even in the linear case, our method can significantly outperform Jalal et al., 2021a for general matrices A, as illustrated in Remark 3 and demonstrated in Appendix E. Please refer to our response to the above Q1 for a detailed explanation.
>
>  - Compared with  Liu & Liu, 2022, first of all, regarding the net performance, our method achieves significantly better results, as demonstrated in various experimental settings. Second, regarding the methods or techniques themselves, while both use generative models, ours are fundamentally different from Liu & Liu, 2022: ours is a sampling method based on diffusion models, while Liu & Liu, 2022 builds on GAN/VAE from an optimization perspective.  It is known that GAN might suffer from unstable training and less diversity due to the adversarial training nature and that VAE/GAN can have large representation errors or biases with inappropriate latent dimensionality and/or mode collapse. By contrast, diffusion models have proven very effective in density estimation and image generation and even outperform the SOTA GAN.
>
> $\bf{Q5}$: As for reproducibility, I hope that the code has been submitted along with the main paper, instead of saying that "Our code will be open-sourced after acceptance".
>
> $\bf{A5}$: Thank you for the comments. We have submitted our code to reproduce the results in the supplementary materials in the rebuttal. Please let us know if there is any problem.

---

### Author Response · Authors · 2022-11-18
**Author response to all reviewers: a summary of the revision**

First of all, we sincerely thank all the reviewers for their time and effort in carefully reading our manuscript and for their valuable feedback, which we believe has significantly improved our work.  In the revised manuscript, we have incorporated all the points of each reviewer. A high-level summary of the main changes in the rebuttal is shown below:

1. We have rephrased Theorem 1 in a clearer and more rigorous form by introducing two assumptions, one is the assumption of orthogonality for the rows of measurement matrix A, and the other is an uninformative assumption of the prior. Verifications of the two assumptions are also provided in Appendix A and B, respectively.

2. We have added results on FFHQ 256x256 super-resolution images for heavily quantized CS with 1-bit, 2-bit, and 3-bit measurements, respectively. The results show that even with such coarse quantization and a small number of (8x) measurements $M\ll N$, high-quality images can still be reconstructed by our proposed QCS-SGM, which demonstrates the superior performance of QCS-SGM with excellent generalization to higher-resolution images.

3. We have added comparisons of different algorithms for multi-bit quantization cases under the fixed-budget setting, which shows that our  QCS-SGM not only is significantly advantageous for 1-bit CS but also remarkably outperforms others in the multi-bits case.

4. We have added all the related works in the references pointed out by the reviewers, i.e., Daras et al. 2022, ISTA-net Zhang 2017, and Neumann networks Gilton 2019. Moreover, we have added more detailed discussions on the difference between these related works and ours, and provide explanations why ours outperforms previous ones. In particular, we would like to highlight three points as follows:

    - QCS-SGM explicitly considers quantization while previous methods, including Jalal et al., 2021a;b, Daras et al. 2022, only focus on linear measurements without quantization. This is the most important reason why QCS-SGM outperforms previous ones. It is worth pointing out that, as shown in our manuscript,  the extension of the linear case to non-linear quantized measurements is far from trivial since it is intractable in general.

    - In calculating the likelihood score of $p(y|x_t)$, previous methods such as Jalal et al., 2021a;b directly compute the score of $p(y|x_0)$ by assuming that $x_t = x_0$, which is apparently not the case for $t>0$ and thus leads to an inaccurate approximation of the likelihood score, thus resulting in performance degradation. In fact, in the original Jalal et al., 2021b, they have to add an additional annealing term to compensate for their inaccurate approximation (otherwise, it does not work well), which is in fact a special case of our generalized results, as illustrated in Remark 3 of our manuscript. This is the second reason why QCS-SGM even outperforms Jalal et al., 2021a;b in the linear case for general matrices A. We have provided a comparison with Jalal et al., 2021a in the linear case on MNIST, as shown in Figure 8 and Figure 9 of Appendix E to illustrate this difference.

    - Both ISTA-net Zhang 2017 and Neumann networks Gilton 2019 are supervised methods that require pairs of data samples $(x,y)$ for training. By contrast, our method is unsupervised and only the data samples $x$ are required for the training. While the supervised approach can achieve good results for specific settings, it is very inflexible since we need to re-train the model once the measurement model is changed, even slightly, e.g., different number M of measurements. However, in real-world applications, we are often confronted with multiple, even possibly infinite, linear measurement models, which makes the supervised approach inflexible.

5. We have also added comparisons with Neumann networks Gilton 2019 under CS (x2) and CS (x8) in the linear case, i.e., without quantization, for both CIfar10 and CelebA datasets, in exactly the same setting as Neumann networks Gilton 2019, in Appendix I (Table 2, Figures 16-19). The results show that QCS-SGM outperforms the Neumann networks Gilton 2019 in all cases. As they did not account for the quantization, it is believed that the performance gap between Neumann networks Gilton 2019 and our QCS-SGM will become much larger in the quantized case, which is the main focus of our work.

6. We have corrected all the typos and revised the manuscript thoroughly.

7. We have submitted our code to reproduce the results in the supplementary material. Please let us know if there is any problem. As pledged before, we will open-source our code upon acceptance.


We sincerely hope that the revision addresses all the concerns and that the reviewers would consider re-evaluating our work. We are happy to provide further clarification if there is any remaining problem. We deeply thank all the reviewers again for their valuable comments.  Individual responses to each reviewer can be found below.

---

### Decision · Program_Chairs · 2023-01-20

**Decision:**

Accept: poster

**Justification For Why Not Higher Score:**

While the paper is generally of high quality and the implementation of the ideas is well done, the reconstruction techniques themselves are a tweak of well-known score-baed generative modeling (SGM).

**Justification For Why Not Lower Score:**

There is consensus that the paper makes interesting contributions, and as such merits acceptance.

**Metareview: Summary, Strengths And Weaknesses:**

The authors introduce a new method for reconstructing images from quantized compressed sensing measurements using score-based generative models; they call their method QCS-SGM. The basic idea is to calculate a quantity called the noise-perturbed pseudo-likelihood score, which can be computed in closed form under certain assumptions. This enables them to implement annealed Langevin Dynamics, leading to a natural reconstruction algorithm. Conceptually, the method extends (and strengthens) previous works in this direction that consider the linear (unquantized) case. Results on a number of standard benchmark datasets confirm the validity of the approach.

Overall, reviews for this paper were positive. While the proposed technique is a combination of well-established ideas, the new pseudo-likelihood calculation is clean and theoretically supports certain algorithmic choices made in a related approach (Jalal et al, 2021). Moreover, the setting of (highly) quantized observations is practically compelling, and the authors do a good job of validating the proposed method on benchmarks.  Concerns were raised about novelty compared to these earlier papers, but I feel that on the whole the paper merits acceptance.

One comment (not raised by any reviewers): the paper does not emphasize this point too much, but it may be helpful to illuminate the role of pre-quantization noise, and how that interacts with the reconstruction procedure. From earlier results in 1-bit CS, it is known that some amount of noise may be necessary for recovery since it provides a "dithering"/regularization effect; however, too much of this may affect the pseudo-likelihood estimates. Consider exploring this tradeoff while preparing future revisions.

**Note From Pc:**

if the above contains the word "oral" or "spotlight" please see: "oral" presentation means -> notable-top-5% and "spotlight" means -> notable-top-25%. As stated in our emails, we are disassociating presentation type from AC recommendations

**Summary Of Ac-Reviewer Meeting:**

N/A